



# Spatiotemporal dynamics and interrelationship between soil moisture and groundwater over the Critical Zone Observatory in the Central Ganga plain, North India

Saroj Kumar Dash[1], Rajiv Sinha[1]

[1]Department of Earth Sciences, Indian Institute of Technology Kanpur, Kanpur 208016, India

*Correspondence to*: Rajiv Sinha (rsinha@iitk.ac.in)

**Abstract.** The understanding of spatiotemporal dynamics of Earth's hydrological components and their controls is critical for efficient water resource management, especially in the agriculture-dominated landscapes. In this study, we utilize the empirical orthogonal function (EOF), random combination, and temporal stability approach on the soil moisture (SM) and depth to groundwater table (DTGT) observations from the Critical Zone Observatory in the Ganga basin to understand their spatiotemporal variability and optimal sampling strategies. Around 91% of the observed DTGT spatial variation are explained by the first two spatial EOF whereas the first five EOFs explains only 67% of the total SM variability. Topography and soil texture (% clay) are considered to be the leading factors that drive the spatial pattern of both the attributes. Furthermore, we noted that four SM sampling locations and two monitoring well, selected randomly can capture the mean spatial variability with an accuracy of 3% $vol/vol$ and 0.90 $mgbl$ (meter below ground level) respectively. Moreover, four temporally stable SM sites and a single observation well are identified, which provide the spatial mean with an absolute error of ±2% $vol/vol$ and 0.36 $mgbl$ respectively. Overall, this study provides an insight to spatiotemporal hydrological controls in an intensively managed landscape and has important implications for water resource management in such regions.

## 1 Introduction

Variability of Earth's hydrological components, integrated over space and time influence the global terrestrial water cycle and sustaining ecosystems (Blume et al., 2009; Taylor et al., 2013). Recent development of the concept of Earth's critical zone (CZ), from treetops to groundwater (Brantley et al., 2016; National Research Council, 2001) and thereby a critical zone observatory (CZO) provides the hydrological basement to understand the spatiotemporal characteristics of surface and sub-surface hydrological processes (Western et al., 2004). Soil moisture and groundwater form two major components in the CZ processes and their interaction controls the exchange of energy and water between land surface and atmosphere, characterizing a highly complex feedback mechanism (Seneviratne et al., 2006; Vereecken et al., 2014; Western et al., 2004). Several meteorological networks and SM stations have been established worldwide (Dorigo et al., 2011), however, local variabilities in in-situ hydro-physical properties such as soil type, topography and precipitation lead to several uncertainties (Broca et al., 2010; Famiglietti et al., 1999). There has also been a profound multiplicative impact of the anthropogenic causal factors such



as the spatially varying groundwater extraction which can lead to alteration of the near surface soil moisture (Miguez-Macho and Fan, 2012; Soylu et al., 2011). Therefore, an understanding the spatiotemporal dynamics of the soil moisture and groundwater level depth and their associated controls is critical to improve the water resource managements, particularly in the agroclimatic domains (Liu et al., 2016).

Several methods exist to characterize the spatiotemporal variability of soil moisture and groundwater level such as the interpolation techniques (Ruybal et al., 2019; Zhang et al., 2019), principal component analysis (de Queiroz et al., 2020; Misi et al., 2018), trend analysis (Joshi et al., 2021; Ye et al., 2019), multiple linear regression (Jia et al., 2020), and singular spectrum analysis (Cao and Zheng, 2016). However, natural complexity and local anthropogenic alteration of near surface components limit the detailed understanding of the hydrological system in a crop water system. On the other hand, the empirical orthogonal function (EOF) analysis effectively analyses large spatiotemporal observations and has been extensively used in many hydrological processes (Joshi and Mohanty, 2010; Meng et al., 2022; Jawson and Niemann, 2007; Yu and Chu, 2010; Yue et al., 2020).

The EOF analysis is a multivariate statistical method, used in decomposing the large space-time datasets into a set of spatial orthogonal functions and temporal expansion coefficients (Hannachi et al., 2007). The EOFs can also be correlated with the regional characteristics to predict major controlling factors (Jawson and Niemann, 2007). Apart from hydrological applications, this method has been extensively applied in other disciplines such as the meteorology (Hannachi et al., 2007; Meher and Das, 2020), earthquakes (Chao and Liau, 2019), and other physical environments (Sauquet et al., 2000). A very recent application of EOF for analysing annual soil moisture observations in the Mongolian plateau showed that a 54% variability could be explained by a single primary mode of decomposition, governing with the groundwater as the greatest influence on it (Meng et al., 2022). Investigation of the physical controlling factors of the Soil Moisture Experiment 2002 (SMEX02) by Joshi and Mohanty (2010) suggested a mixed effect of the rainfall, topography and soil texture on the major EOFs. Application of the EOF analysis to monthly observation of groundwater table depth from 67 monitoring well in parts of Northwest China found the evaporation and temperature as the major drivers of the temporal variability (Yue et al., 2020). Another important aspect of the in-situ spatiotemporal hydrological data are the optimization of the sampling locations representing the mean catchment variability using the (a) random combination analysis and (b) temporal stability approach. Estimating the number of required samples (NRS) for mean variability through random combination approach has been explored by many authors (Brocca et al., 2010, 2012; Chen et al., 2016; Zhao et al., 2013). For instance, Brocca et al. (2012) and Singh et al. (2019) selected 2 and 10 random sites within an area of $100\,km^2$ in Upper Tiber River, central Italy, and Mahanadi River basin in India respectively to estimate the catchment mean with a 2% error. Surprisingly, studies pertaining to the implementation of this approach in groundwater measurements are almost non-existent.

The concept of temporal stability was first introduced by Vachaud et al. (1985) for the SM dynamics and its implementation to estimate time-invariant characteristics of sampling locations has been documented in the literature (Li and Shao, 2014; Sur et al., 2013; Wang et al., 2013). Again, most of the work conducted in the past used this concept in depicting the representative





sampling site for SM whereas very few studies addressed the spatial groundwater variability (Wang et al., 2018; Xu et al., 2015).

Therefore, the present study was carried out with the aim of addressing the following research questions: (a) understanding the spatiotemporal dynamics of SM and groundwater level and its influence through EOF analysis, (b) exploring the concept of random combination and temporal stability analysis for the mean catchment SM and groundwater variability. The study has been conducted over an agricultural critical zone observatory (CZO) located in the central Ganga plain in North India.

## 2 Study region and datasets

### 2.1 Study area

Soil moisture and ground water measurements were carried out in an agriculture-dominated Critical Zone Observatory (CZO) in the Pandu basin, a small plains-fed tributary of the Ganga River (Fig. 1). The CZO here is referred to as the HEART (Heterogeneous Ecosystem of an Agro Rural Terrain) CZO with an area of ~21.5 km$^2$ and was established in 2016 (Gupta et al., 2019) in the rural parts of the Ganga plain. The predominant land use in this region is the agriculture, constituting more

than 90% of the total area. The elevation in the HEART CZO ranges from 126 to 143 m above mean sea level. This region falls in sub-humid climatic regime with the observed average annual maximum and minimum temperatures of 42°C and 8.6°C, respectively. Mean annual rainfall in the CZO is approximately 821.9 mm where the monsoonal rainfall (June-September) contributes more than 90%. Soil texture is predominantly sandy loam and loam and the major crops grown are rice and wheat in Kharif (July-October) and Rabi (October-March) season, respectively.

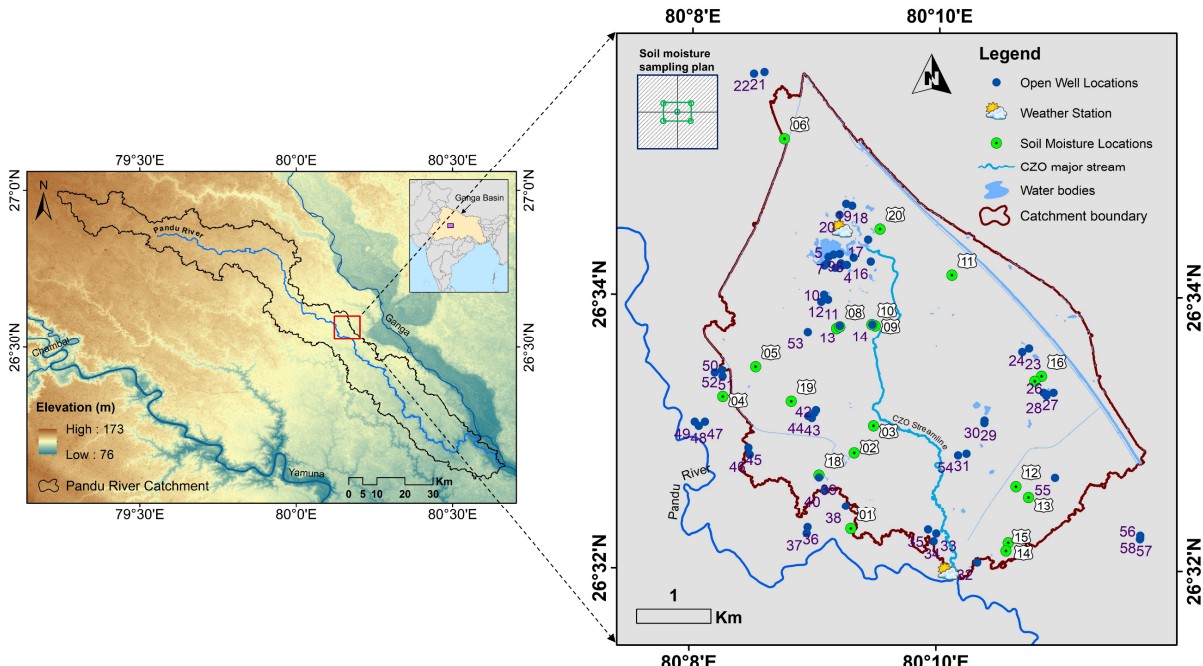



**Figure 1: Location map of the study area, the Critical Zone Observatory in the Pandu basin. The study region is located within the Ganga-Yamuna interfluve. The soil moisture and the ground water monitoring locations along with the in-situ weather station are shown in right. Location index, 06, 11 and 20 of soil moisture are irrigated with surface water bodies (river and canal) whereas cropping at location index 03, 04 and 19 mostly produced using both surface and groundwater sources. The remaining locations are**
**irrigated using groundwater sources. The CZO, shown in the right also includes a sampling scheme (not to scale) adopted for the soil moisture measurements (inset on the upper-left corner). The green circles of the sampling scheme represent the Theta probe measurement in an agricultural field.**

## 2.2 SM measurements

Based on the heterogeneity in agricultural water use, soil texture and topography, a total of 20 different sites were selected for
SM monitoring. The measurements were carried out manually at each location. All measurement locations are placed within the agricultural fields which use different sources of water such as the canal, groundwater, and river (Fig. 1). A total of 62 days (from September 2017 to December 2019) of sampling was done to capture the inter-annual variability. These periods excluded the summer months, when the fields are completely dry and days of surface ponding because of ample precipitation. Furthermore, multiple linear regression was used to approximate the unmeasured sampling locations on a given day based on
its proximity to the nearest sampling location and the overall annual cropping patterns. Also, the field campaign was mostly conducted in early morning based on the dates of descending pass of Soil Moisture Active Passive (SMAP) satellite over the study region.

A sampling scheme of five measurements in and around the centre of each agricultural field was chosen to represent the mean SM of the site (Fig. 1). The sampling was conducted manually using a handheld impedance based ML3 ThetaProbe SM sensor
(Delta-T Devices, Cambridge, England) with a reported accuracy of ±1%. The sensor was also calibrated prior to field campaign using adequate soil samples from the in-situ locations. Accordingly, 37 soil samples were tested in the laboratory using the gravimetric measurements for establishing the calibration equation although Kaleita et al. (2005) showed that 20 samples are adequate to calibrate the ThetaProbe. Calibration coefficients were derived by comparing the gravimetric measurements of the water content along with the simultaneous measurement through the sensor (Delta-T readings) from a
saturated to wilting state of soil samples Eq. (1).

$$SM_{actual} = 0.9557 \times SM_{observation\,(ThetaProbe)} + 0.31 \qquad (1)$$

## 2.3 Groundwater measurements

A total of 58 open observation wells throughout the CZO have been monitored to estimate the spatiotemporal variability of the groundwater level in this region (Fig. 1). The ground water level (GWL) recorder (Virtual Hydromet, Uttarakhand, India)
was used to measure the depth to groundwater table (DTGT) at each well at a biweekly interval. The monitoring period was selected to be same as the SM campaign to observe the integrated spatiotemporal variability. The measurements were mostly conducted in 2017 and 2019 with very few observations during 2018 constituting a total of 22 monitoring days. The annual maximum and minimum value for the groundwater level are found to be -0.06 m and -8.32 m respectively (negative value is indicative of the depth below the surface).





In addition, meteorological data from in-situ automatic weather stations, elevation from shuttle radar topographic mission (Farr et al., 2007), and soil texture information from SoilGrids (Poggio et al., 2021) were used for characterization of the in-situ attributes of the observation locations.

## 3 Methods

### 3.1 Statistical analysis

Both SM and groundwater data were analysed to compute the temporal mean and coefficient of variation. If $\theta_{ijk}$ is the in-situ SM measurement or groundwater data observed at point $i$, sampling location $j$ and sampling day $k$, then,

$$Sampling\ mean,\ \bar{\theta}_{jk} = \frac{1}{N_p}\sum_{i=1}^{N_p}\theta_{ijk} \qquad (2)$$

where, $N_p$, is the number of point measurements within an agricultural field, $j$, and sampling day, $k$. Using this approach, the spatial mean of both the SM and DTGT in CZO for each sampling day, $\bar{\theta}_k$, and temporal mean for each sampling location, $\bar{\theta}_j$,

was calculated as follows:

$$\bar{\theta}_k = \frac{1}{N}\sum_{j=1}^{N}\bar{\theta}_{jk} \qquad (3)$$

$$\bar{\theta}_j = \frac{1}{M}\sum_{k=1}^{M}\bar{\theta}_{jk} \qquad (4)$$

where, N is the total number of sites and M is the total number of sampling days in the campaign.

The coefficient of variation ($CV$) for $k^{th}$ sampling day in space, $CV_k$ was computed as:

$$CV_k = \frac{\sigma_k}{\bar{\theta}_k} = \frac{\sqrt{\frac{1}{N-1}\sum_{j=1}^{N}(\bar{\theta}_{jk}-\bar{\theta}_k)^2}}{\bar{\theta}_k} \qquad (5)$$

where, $\sigma_k$ is the spatial standard deviation. Similarly, the $CV_j$ for $j^{th}$ sampling location can be defined analogously.

### 3.2 Empirical orthogonal function (EOF) analysis

This study employed the EOF analysis to decompose the spatiotemporal SM and DTGT data into a set of spatial empirical orthogonal functions (EOFs) and temporal expansion coefficients (ECs). The maximum variation in the datasets can be

explained by first few EOFs/ECs. The EOF analysis for a space-time dataset is described as below.

The spatial anomaly matrix (X) for a given space-time datasets need to be computed as below:

$$X = \begin{bmatrix} x_{11} & \cdots & x_{1M} \\ \vdots & \ddots & \vdots \\ x_{N1} & \cdots & x_{NM} \end{bmatrix} \qquad (6)$$

where, $x_{jk} = \bar{\theta}_{jk} - \bar{\theta}_k$ is the individual spatial anomaly for the $j^{th}$ location and $k^{th}$ sampling day and $\bar{\theta}_{jk}$ is the sampling mean at each location and $\bar{\theta}_k$ is the catchment spatial mean. N is the total number of sites and M is the total number of sampling

days.

Then, the covariance matrix $C$ can be calculated as:





$$C = \frac{1}{M} X \cdot X^T \tag{7}$$

where, the superscript $T$ indicates the transpose of the matrix.

The EOFs and ECs can be solved by computing the eigen vectors and eigen values of $C$, satisfying the following relation:


$$C \times V = V \times E \tag{8}$$

where, V is the eigen vectors having $N \times N$ dimension and E is the diagonal matrix of eigen values such as:

$$E = \begin{bmatrix} \lambda_1 & \cdots & 0 \\ \vdots & \ddots & \vdots \\ 0 & \cdots & \lambda_N \end{bmatrix} \tag{9}$$

The ECs (F) and the percentage of individual EOF variance ($EV_k$) can further be calculated as:

$$F = V^T \times X \tag{10}$$


$$EV_k = \frac{\lambda_k}{\sum_{i=1}^{N} \lambda_i} \times 100 \tag{11}$$

This process transforms the datasets into several spatial EOFs that explain the decreasing trend of variability in a multidimensional space. In addition, the relation between the leading EOFs and the in-situ physical parameters has been assessed in this study using the Pearson's correlation coefficient for both the SM and the DTGT.

### 3.3 Optimal sampling design

### 3.3.1 Random combination approach for determining number of required samples (NRS)

In this study, we applied the random combination approach based on the bootstrap technique to determine the NRS, characterizing the mean catchment dynamics for SM and DTGT (Brocca et al., 2010, 2012; Wang et al., 2008, Zhao et al., 2013). This method is independent of any assumption on sampling statistical distribution (Wang et al., 2008). Fig. 2 describes the schematic workflow of the random combination approach for the space-time data. More details of the workflows of random

combination approach have been discussed by Wang et al. (2008) and Brocca et al. (2012). This approach resulted in a time series of the mean and the standard deviation of each randomly combined series which were evaluated against benchmark value using the root mean square difference (RMSD) and coefficient of determination ($R^2$) values.


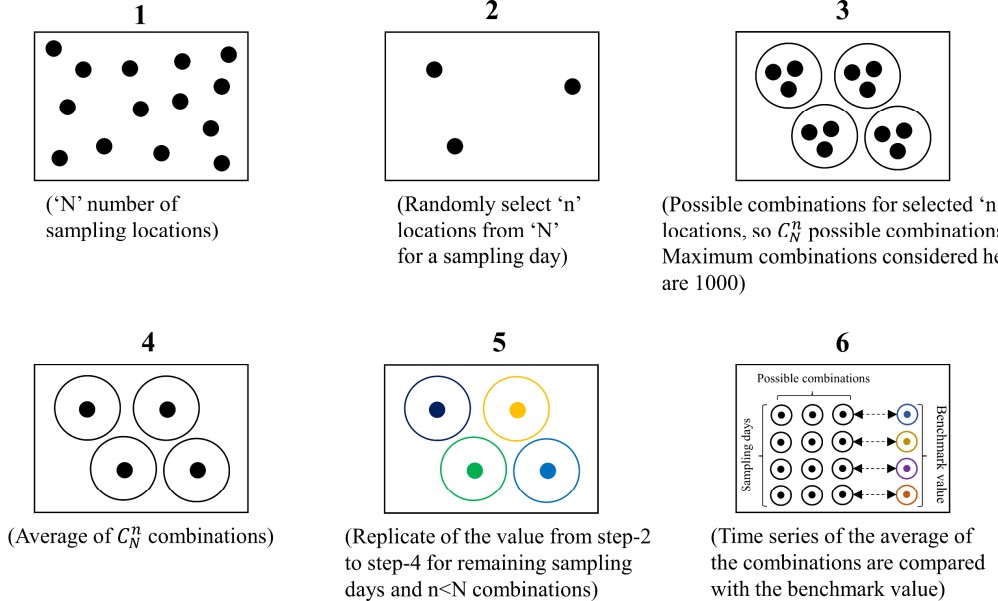

**Figure 2: Sequence of operation for the random combination approach**

### 3.3.2 Temporal stability analysis

The NRS obtained above only computes the number of sampling locations (in-situ points for SM and open wells for groundwater) considered randomly for mean SM or groundwater variability for a region. However, identification of a representative site/open well in space to estimate the absolute mean value (within a predefined accuracy), can be achieved through temporal stability analysis, proposed by Vachaud et al. (1985). Temporal stability analysis can be mathematically explained as below.

For a sampling location $j$, sampling day $k$, the relative difference, $\delta_{jk}$ was calculated as:

$$\delta_{jk} = \frac{\bar{\theta}_{jk} - \bar{\theta}_k}{\bar{\theta}_k} \tag{12}$$

The mean relative difference, $\bar{\delta}_j$ and the standard deviation, $\sigma(\delta_j)$ were calculated for each sampling locations as per the following expressions:

$$\bar{\delta}_j = \frac{1}{M}\sum_{k=1}^{M}\delta_{jk} \tag{13}$$

$$\sigma(\delta_j) = \sqrt{\frac{1}{M-1}\sum_{k=1}^{M}\left(\delta_{jk} - \bar{\delta}_j\right)^2} \tag{14}$$

The $\bar{\delta}_j$ of a sampling location helps to identify the wetness and dryness compared to the areal mean. This can be identified with a low value of $|\bar{\delta}_j|$ and $\sigma(\delta_j)$. In addition, a single metric called the Index of Stability (ITS) which combines both ($|\bar{\delta}_j|$ and $\sigma(\delta_j)$) can also be used for better evaluation of the sampling locations (Jacobs et al., 2004; Zhao et al., 2020b; Zhu et al., 2020). This is calculated for each site using the following expression:





$$ITS = \left( \bar{\delta_j}^2 + \sigma\left(\delta_j\right)^2 \right)^{1/2} \qquad (15)$$

A major advantage of determining the time stable location through ITS is that both relative difference and its variances are taken into consideration. Jacobs et al. (2004) suggested that the location having a lowest value of ITS can be considered as a temporally stable location.

## 4 Results and analysis

### 4.1 Spatiotemporal mean and variability

Temporal dynamics of soil moisture and the DTGT are shown in Fig. 3, that represents the overall seasonal behaviour of surface dryness/wetness and the subsurface groundwater level. Fig. 3a shows the spatial variability of field mean SM with ±1 standard deviation for each sampling day in addition to the daily rainfall observed at the in-situ weather station. Spatial mean SM values for the entire field campaign ranges from 10% (vol/vol) to 43% (vol/vol). The SM values show a high dependence on the prevailing rainfall pattern in the region. This is primarily observed during the monsoon period (July-September) of the year. The standard deviation during the monsoon period shows a low value because of low spatial variability, while the non-monsoon periods show a high value of standard deviation.

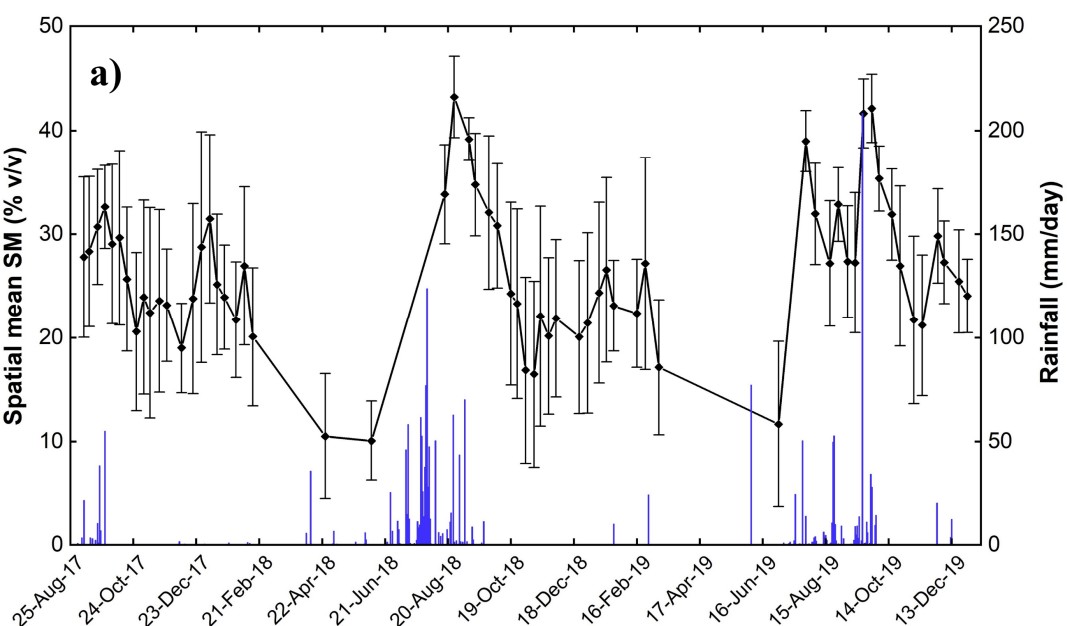



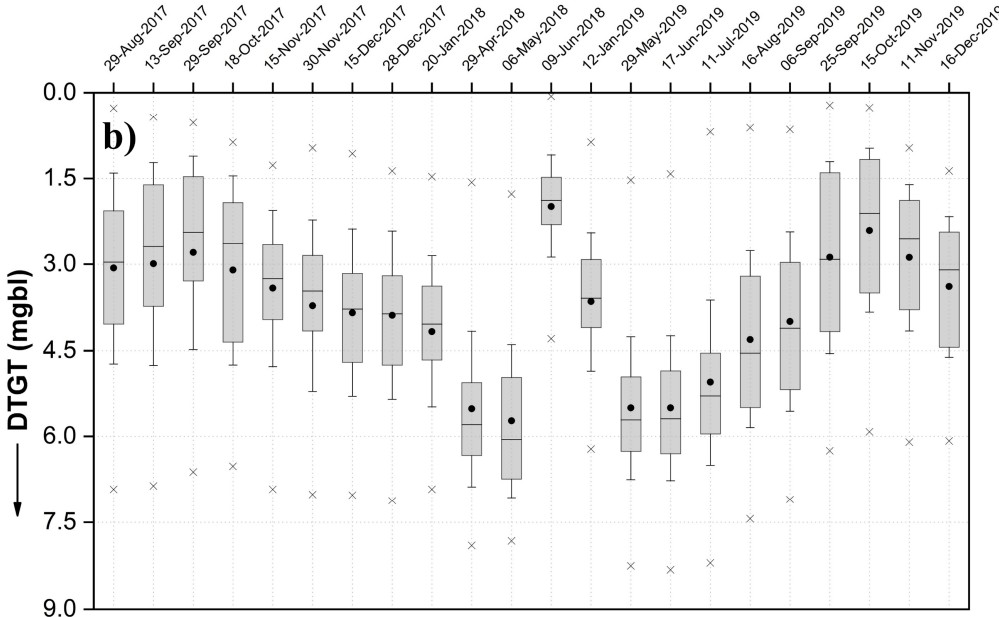


**Figure 3: a) Temporal variation of spatial mean soil moisture with ±1 standard deviation. The blue bars are the daily rainfall observed at the in-situ weather station. b) Box-Whisker plot showing the date-wise variation of depth to groundwater table (DTGT) in CZO. The top and bottom edge of box represent the data within interquartile range (25%-75%), whereas the inside horizontal line shows the median value. Filled circle, inside the box represents the mean and the whisker shows the standard deviation (±1)**
**from mean value. The cross marks at both the end of the box shows the minimum and maximum value.**

Fig. 3b shows the diurnal variation of the DTGT in the CZO for the specific dates of observation. The variation in the mean level of groundwater depth follows the seasonality of precipitation in the study region. The interquartile range for all sampling days shows a significant range of variability including the monsoon period (June-September), when there is ample precipitation. Also, the mean DTGT shows a decreasing pattern from pre-monsoon (March-May) to monsoon (June-

September) and an increase during the post-monsoon period (October-December) and winter (January-February). A sharp change in the mean DTGT on 9[th] June 2018 is attributed to the early and infrequent monsoon rainfall that resulted in rise of the groundwater level.

The temporal mean of both the datasets are presented in Fig. 4. Soil moisture data shows a wide annual variation because of natural and human-induced water fluxes whereas the groundwater level primarily depends on infiltration and abstractions for

irrigation. Temporal variability for SM was found to be 20-31% vol/vol. Low/high temporal means of a particular SM location indicate the drier/wetter characteristics of the sampling point. The measurement location L-06 has the highest value of temporal mean, and therefore, minimum variability ($\sigma_j = 5.96\%\ vol/vol$). It is noted that the lowest temporal mean (19.89% vol/vol) is observed at location, L-14, and the highest standard deviation ($\sigma_j = 11.92\%\ vol/vol$) is at location, L-15; both sites are located downstream of the catchment. Temporal mean of DTGT ranges from 1.09-6.59 $mgbl$ with a standard deviation of (±)

0.39-1.93 $mgbl$. Monitoring wells close to each other have similar characteristics, and therefore, show similar mean values





and standard deviation. Also, a similar trend in the depth to groundwater table for all monitoring wells was observed which suggests that the rate of change in the water pressure for a particular sampling day is same for all the observation wells.

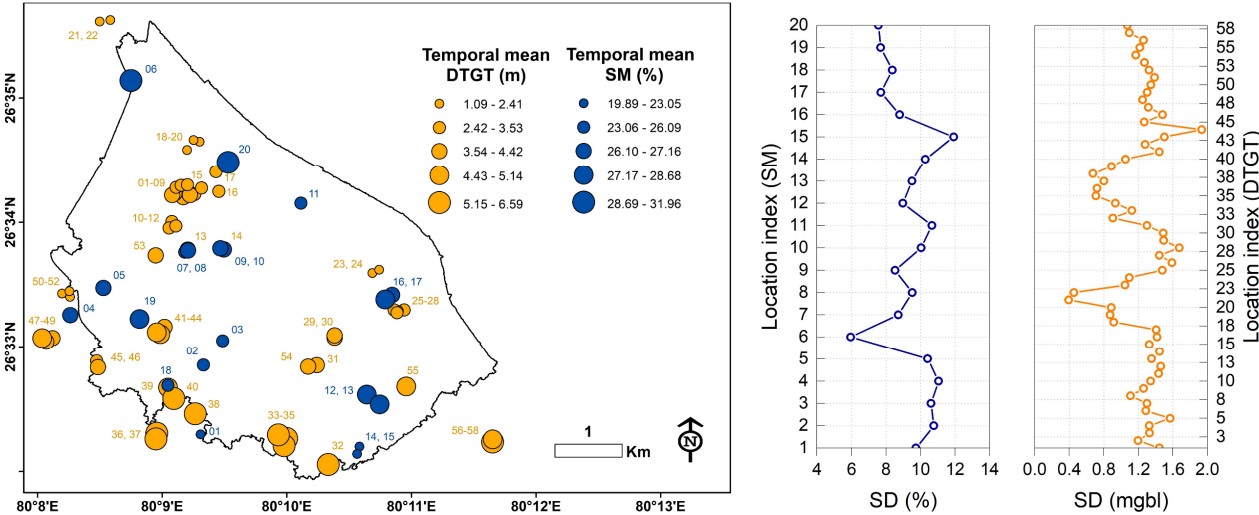

**Figure 4: Spatial pattern of the temporal mean of soil moisture (SM) and depth to groundwater level (DTGT) for each location. The**
**size of the bubble represents the temporal mean value for the SM (blue) and the DTGT (orange). Locations of all the locations of SM (text in blue) and the DTGT (text in orange) are shown for better interpretation. The temporal standard deviation (±1) of the SM (blue line) and the DTGT (orange line) for all the measurement locations are shown in the right.**

### 4.2 Statistical analysis of the SM and DTGT

The statistical analysis has been carried out to investigate the spatiotemporal variability in both the SM and the groundwater
level depth. Considering the statistics of higher order moments, i.e., standard deviation (SD) and coefficient of variation (CV), the spatial variability of both the datasets are presented in Fig. 5. The variation in CV allows to compare the measurement variability across different spatial and temporal scales. On the spatial scale, the spatial $CV_k$ of observed SM was found to be fairly high (maximum 0.69) during the dry period (1$^{st}$ July 2019) and quite low (0.05) during the monsoon (9$^{th}$ September 2018) (Fig. 5a). On an average, the spatial variability, $CV_k$ was found to be 0.28 and can be ascribed not only to the number of
measurements points but also to the physical water input (watering by farmers) at the sampling site before measurements. However, temporal coefficient of variation ($CV_j$) varies from 0.19 to 0.54. Further, the $CV_k$ for the spatial DTGT ranges from 0.23 to 0.60 with an average of 0.4 (Fig. 5b). Although higher order spatial DTGT variability has been observed at CZO, but this refers to the number and seasonality of the selected sampling days (22 in total). A close examination of the $CV_j$ on SM and DTGT observation data shows that the average value in both cases is the same i.e., 0.36, suggesting a consistent seasonal
variability of both.

The observed decreasing trend between $\bar{\theta}_k$ and $CV_k$ was embedded (Fig. 5a) and was fitted with the analytical exponential relationship (Eq. 16), as usually demonstrated in the past studies (Brocca et al., 2010, 2012). Here, we also applied the same relationship to the DTGT time series, and the result shows a trend similar to the SM (Fig. 5b). The fitting parameters along





with the determination coefficient ($R^2$) are shown in Fig. 5.

$$CV_k = a. e^{-b.\bar{\theta}_k} \tag{16}$$


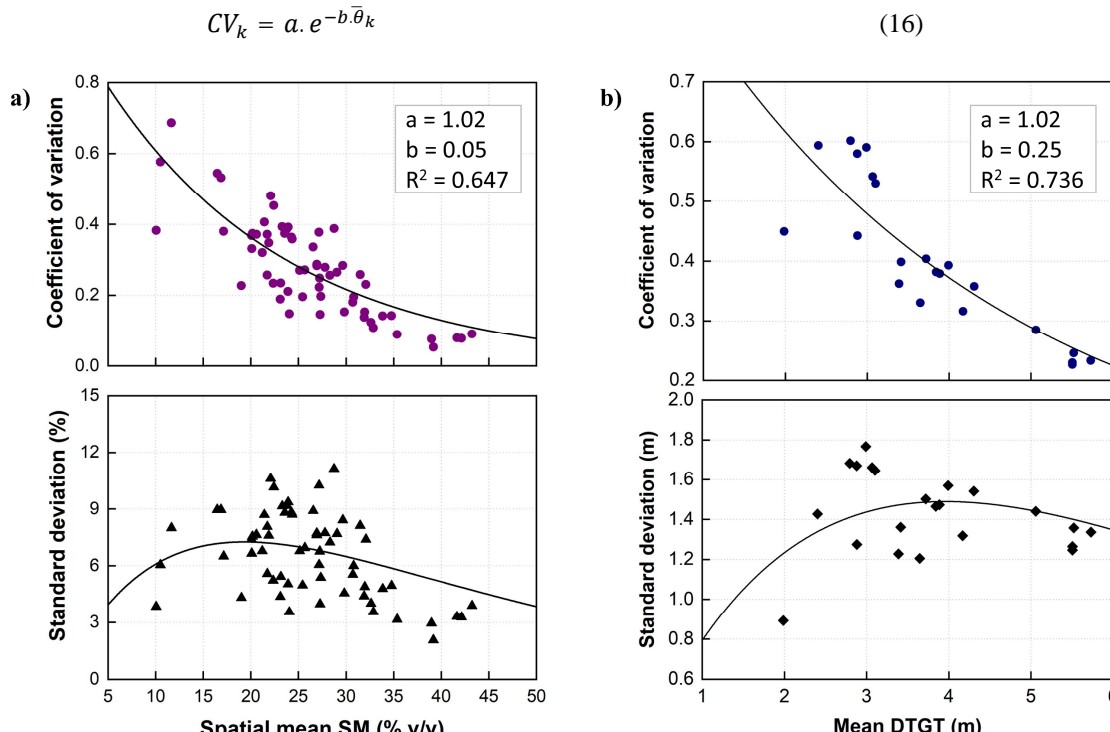

**Figure 5: Relationship between spatial mean a) soil moisture and b) DTGT with standard deviation and coefficient of variation. The solid lines are fitted through an exponential relationship between the mean-coefficient of variation ($CV_k = a. e^{-b.\bar{\theta}_k}$) and the mean-standard deviation ($\sigma_k = a. e^{-b.\bar{\theta}_k}. \bar{\theta}_k$). The fitted parameters of $CV_k$- $\bar{\theta}_k$ relationships are shown within the box. The fitting between $\bar{\theta}_k$-$\sigma_k$ is based on the parameters derived from $CV_k$-$\bar{\theta}_k$ relations.**


Fig. 5 also indicates the relationships of between the $\bar{\theta}_k$ and the $\sigma_k$ for both the observational datasets. The trend of $\sigma_k$ with the mean SM shows that the variability is high for low ($< 19\%$) soil moisture values and vice versa i.e., low variability in the monsoon period when the soil moisture values are high. Similar interpretations can be made for the groundwater level data, where higher variability of mean DTGT is observed for $< 4m$ below ground level.

**4.3 EOF analysis of soil moisture and groundwater level**

Spatial anomalies of DTGT and SM are subjected to the EOF analysis, yielding 58 and 20 pairs of EOFs/ECs respectively. The EOFs and ECs jointly explain the total spatiotemporal variability of the corresponding earth observation attributes. Fig. 6a-b shows the spatial distribution of the first two EOFs for the SM and DTGT observation. The DTGT spatial distribution of the EOF1 shows positive values ranging from 0.1 to 0.3 towards the downstream portion of the CZO, suggesting a deeper

groundwater level compared to the shallower level at the upstream region with negative EOFs (up to -0.3). At the same time, EOF1 of SM shows high values close to the water bodies with negative values at the downstream end of the catchment. This suggests a drier surface at the downstream, compared to the upstream wetter area. The EOF2 spatial distribution on both the





SM and DTGT shows an increase in positive values compared to EOF1. An overall examination of the spatial patterns of EOF1 and EOF2 reveals a non-uniform variability of both attributes throughout the CZO.

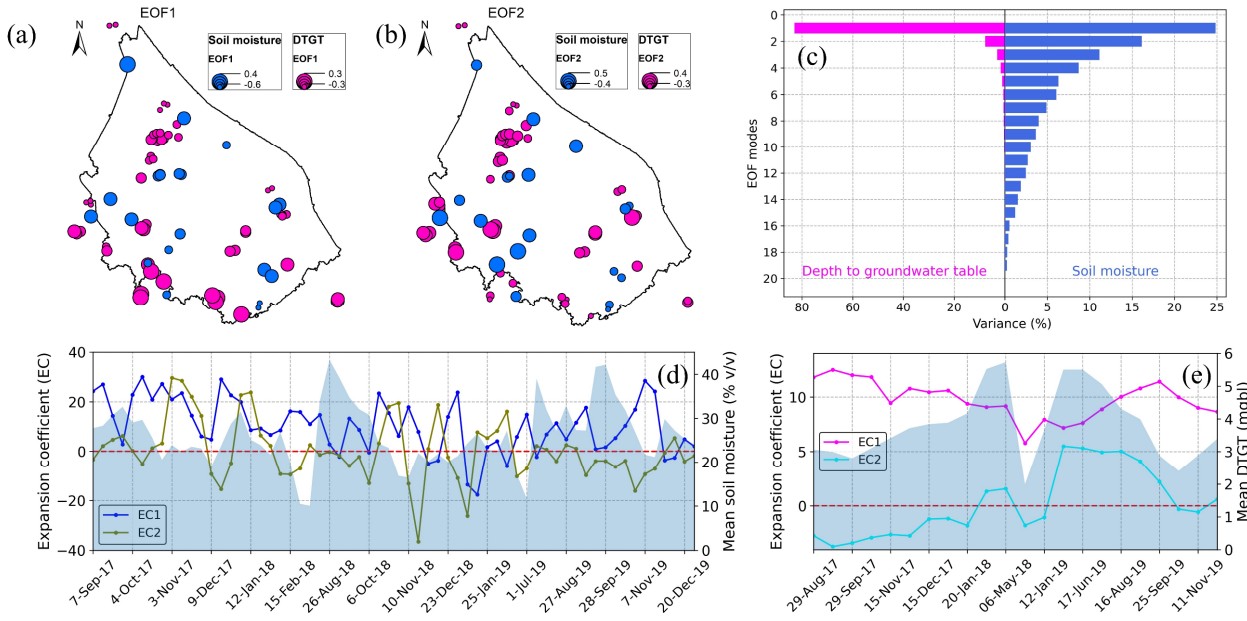

**Figure 6: Spatial representation of the first EOF (a) and the second EOF (b), generated from the spatial anomalies of the corresponding datasets for the CZO, (c) the overall variances of the first 20 EOFs of soil moisture and DTGT, (d) timeseries of the first two ECs for soil moisture. (e) timeseries of the first two ECs for DTGT. The light blue shade in (d) and (e) indicates the time series of actual spatial mean SM and DTGT respectively.**

The spatial variance of each EOF/PC pair is shown in Fig. 6c, where the first twenty pairs of each EOFs are presented which explains almost 100% of the total spatial variability. It is observed that the first two EOFs of the DTGT explain about 91% of the total variability whereas the EOF1 and EOF2 of the SM represent the total variability of only 41% and the first five EOFs of the SM decomposition have a cumulative variance of 67%. This suggests a higher degree of variability of SM compared to DTGT in the CZO.

Fig. 6d and Fig. 6e show the temporal variability of the first two ECs along with the mean variability of SM and DTGT respectively. Both positive and negative EC extremes are observed during the early post-monsoon phase, indicating a greater variability of SM during monsoon-post-monsoon transition period (Fig. 6d). However, some extreme values of EC are observed during the harvesting period of rice and beginning of wheat crop (December-January) in the study region. This suggests an overall similar cropping pattern in most of the crop fields during this period. This also coincides with the period of annual minimum air temperature which influence the SM variability to a great extent. Fig. 6e shows the major lower extremes of the temporal DTGT variability observed on the mid-June of 2018, when the water table lies near to the ground surface because of the infrequent rainfall in the early-monsoon phase. This may also be attributed because of the extremely limited observation in the year 2018. The uppermost value of the EC1 is observed during the peak of the monsoon period



(August-September) of the study region. The EC1 is also deprived of any of the negative weights whereas the second principal
component (EC2) closely follows the observed temporal DTGT pattern of the region with extreme negative weights during
the monsoon in the year 2017.

## 4.4 Drivers of spatiotemporal SM and DTGT dynamics

In this study, we characterize the association of the spatial EOFs with the in-situ geophysical properties such as the altitude,
slope, topographic wetness index (TWI) by utilizing a simple correlation analysis. The TWI is a frequently used terrain index
defined as $\ln(a/\tan\beta)$ where $a$ is the local upslope area draining through a certain point per unit contour length and $\tan\beta$ is
the local slope (Beven and Kirkby, 1979; Sörensen et al., 2006). The TWI influences the spatial distribution of soil moisture
and groundwater level to a significant extent (Alikhanov et al., 2021; Chaplot and Walter, 2003). Fig. 7 presents the correlation
among the first five EOFs with the in-situ time-invariant properties. It is observed that elevation, slope, silt, and clay fraction
are moderately correlated with the primary EOF pattern of spatial SM dynamics, indicating that the variability is related to a
mixed effect of all these attributes (Fig. 7a). In addition, a strong correlation of clay fraction with the spatial SM is observed
in the EOF3 spatial pattern. On the other hand, the sand fraction and TWI seem to play very limited role in the prime spatial
variation of the SM, however, the rest of the EOFs show a good relationship with it. This suggest that the SM variability is
primarily controlled by clay fraction and topography but there is a mutual control by other in-situ physical attributes.

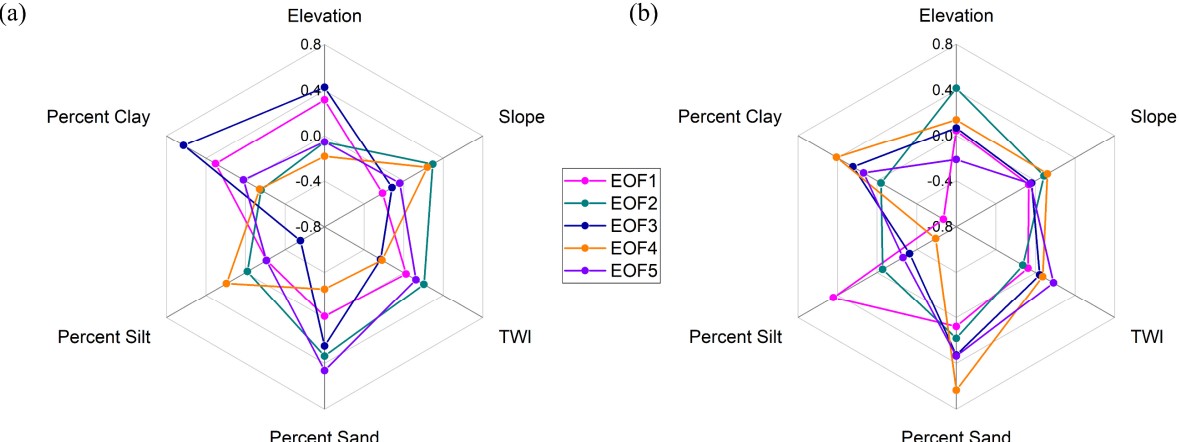

**Figure 7: Relationship between the first five spatial EOFs of soil moisture (a) and DTGT (b) with the in-situ time-invariant**
**geophysical characteristics of the CZO**

The relationship between the EOFs of DTGT spatiotemporal dynamics with the above variables are illustrated in Fig. 7b. In
this case, the relationship of EOF1 distribution suggests that clay distribution has a great control on the groundwater table
variation followed by silt percentage. The sand fraction also has a minor role in controlling the regional water table variation
in the CZO when considering its relationship with rest of the EOFs. Interestingly, elevation although has an extremely low





relation with the main EOF1 distribution of DTGT, this plays a significant role in the spatial variability as observed in the EOF2 pattern.

**4.5 Random combination: temporal evolution of spatial mean**

**4.5.1 SM evaluation**

In this study, the estimation of number of required samples ($NRS_{SM}$) to approach the spatial mean on SM time series was carried out using random combination analysis (62 days and 20 location). The analysis was performed with randomly selected points up to $N - 1$ number of sample locations, where $N$ is the total number of sampling locations (here N=20). The maximum number of combinations or replicates was set to 1000 ($N_r$=1000) to minimize the computation time (Zhao et al., 2013).
Results on the performance statistics of both the SM and the DTGT random combinations are presented in Fig. 8. A single
random SM site can approach to the CZO spatial mean SM value with an accuracy of 6.7% vol/vol with a fairly low determination coefficient of 0.57 (Fig. 8a). Therefore, the selection of a single random site is not suited to capture the temporal pattern in the study region, which was expected because of a large heterogeneity of volumetric water content in the surface soil attributed to anthropogenic factors. The RMSE value decreases and the $R^2$ increases gradually, as the sampling size becomes larger. Moreover, to capture the spatial mean surface wetness with a 3% vol/vol, very limited number of sampling
locations (i.e., 4) are needed throughout the CZO and this can have a $R^2$ value of 0.86. These values improve a lot when the sample size is 11, selected randomly, producing the spatial mean determination coefficient of 0.97 with a RMSD of ±1%, whereas the number of random samples become 6, when the desired error will be ±2%.

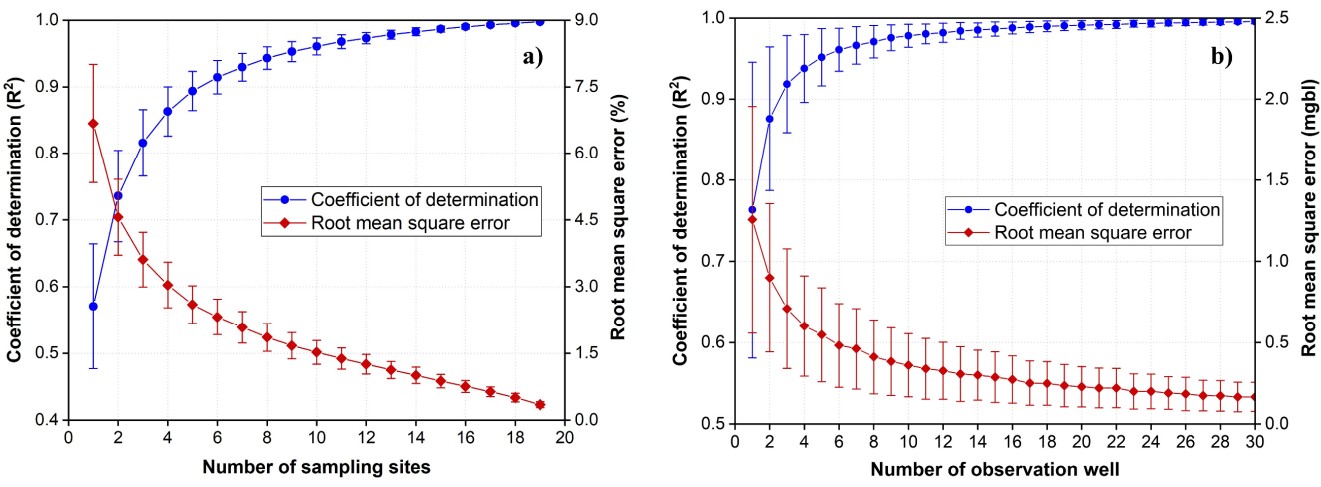

**Figure 8: Coefficient of determination (R2) and root-mean-square-error (RMSE) between the time series of CZO spatial means and**
**the mean value obtained by randomly selecting sampling sites for (a) soil moisture and (b) DTGT. A maximum of 30 random observation wells are shown here to estimate the mean DTGT variation within the study area. The bound indicates ±1 standard deviation corresponding to the mean value.**



### 4.5.2 Groundwater table depth

Similar to the SM dataset, the DTGT data were also analyzed to determine the number of required samples for mean DTGT (NRS$_{DTGT}$) and Fig. 8b represents the performance statistics of the random combination analysis on the DTGT data considering 1000 iterations for each combination. The analysis here was produced considering 57 (here N = 58) number of possible combinations over 22 sampling days. The results shown here are the averages of $R^2$ and the RMSD of the time series between the benchmark time series and the time series obtained by averaging the combinations of randomly selected observation well

in the study area. As compared to SM performance statistics, the performance in the DTGT shows a higher order of $R^2$ and RMSE. It is observed that a single random monitoring well can approach the spatial mean value with evaluation statistics, $R^2$ = 0.76 and RMSE = 1.26 $mgbl$. The $R^2$ value significantly increases when the NRS$_{DTGT}$ = 2. This means that at least two randomly selected DTGT observation wells can capture the spatial mean DTGT in the study region with a $R^2$ of 0.88 and RMSE of 0.90 $mgbl$. The mean error becomes 0.5 $mgbl$, when the NRS$_{DTGT}$ = 5 to 7. Furthermore, more than six observation

wells do not show any significant improvement in the $R^2$ value, but the RMSE seems to be gradually lower as the number of observation wells increases.

### 4.6 Spatial correlation between sampling days

To analyse the similarity in the spatial mean between the sampling days, the Spearman rank correlation coefficients, R, has been computed and presented in Fig. 9a and 9b for SM and DTGT, respectively. Significant correlation was observed for most

of the SM sampling days in 2017 and November 2019. Remaining days are found to be sparsely correlated with the days next to it. Nevertheless, the spatial correlation coefficient depends on the sampling days taken into consideration along with the variability of the individual agricultural fields. Fig. 9a also illustrates a significant correlation observed between 2017 and 2018 towards the end of post-monsoon, when they are separated by one year. Irrigation effects on the sowing of wheat crops during early November 2019 significantly correlate to the monsoon pattern of the previous years. A negative correlation (-0.8

> R > -1) of SM measurements on 11-Jan-2019 was observed for the sampling days in 2017 monsoon months and this lasts up to 30-Jan-2018. Fig. 9b shows the Spearman rank correlation coefficient among different DTGT monitoring dates and interestingly all the observed days are significantly correlated (p<0.01), indicating a strong temporal persistence for all measurements. In addition, significantly stronger correlation in DTGT was observed among sampling days in 2017 and 2018. This suggests a similar variability of DTGT during August-December of each year in the study region with a spatial scale, similar to CZO.

similar to CZO.





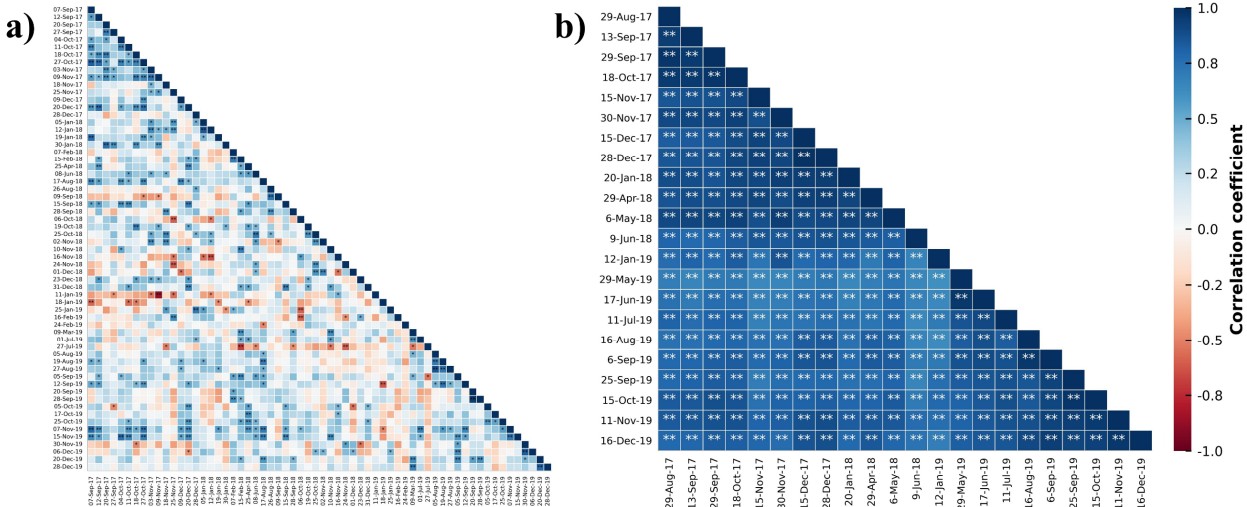

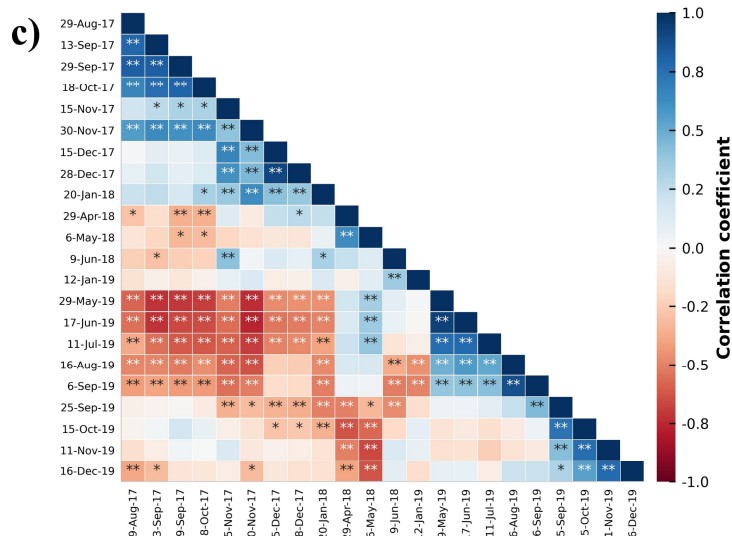

**Figure 9: Spearman rank correlation coefficient of the observed values during the measurement campaigns of a) soil moisture, b) DTGT and c) groundwater storage anomaly. The correlation coefficients are shown as the triangle where the value is represented as generic boxes identified by $i_{th}$ row and $j_{th}$ column and each of the row/column are the sampling dates of the corresponding field campaign. \* Indicates significance at $p < 0.05$, \*\* Indicates significance at $p < 0.01$**

In addition to correlation coefficients of individual DTGT sampling days as discussed above, we also computed the correlation of the groundwater storage anomalies (GWSA), illustrated in Fig. 9c, which shows a wide range of variation. The groundwater storage anomalies are calculated for each of the 58 monitoring wells and for every sampling day. Initially, the sign of DTGT is reversed followed by subtracting long-term mean values from its individual values for each well to get the groundwater level anomaly (Bhanja et al., 2017; Li et al., 2015). The GWSA values are calculated further by multiplying the specific yield (12%





for CZO, CGWB report 2008-2009) with the groundwater level anomaly. Significant correlation (p<0.01) was observed during monsoon of 2017. Also, a stronger negative correlation was found between the pre-monsoon and monsoon months of 2019

and 2017. On an average, both the datasets (SM and DTGT) are well correlated during the monsoon periods, when uniform variability occurs due to rainfall.

### 4.7 Temporal stability analysis

### 4.7.1 Rank based on relative difference

The rank ordered mean relative difference ($\bar{\delta}_j$) and the corresponding standard deviation ($\sigma(\delta_j)$) are presented in Fig. 10a and

Fig. 10b for the entire DTGT and SM data, respectively. The $\bar{\delta}_j$ for SM ranges from -26% to 24%, while the corresponding $\sigma(\delta_j)$ varies from 19% to 36% with a mean variation of 28% (Fig. 10a). The lower number of sampling days, spatial heterogeneity along with the time of crop watering seems to be an influencing parameter to the wide variation of both the parameters ($\bar{\delta}_j$ and $\sigma(\delta_j)$) in CZO. The range of $\bar{\delta}_j$ values for DTGT is quite large varying from -71% to 82% with a $\sigma(\delta_j)$ from 6%-39% (Fig. 10b).

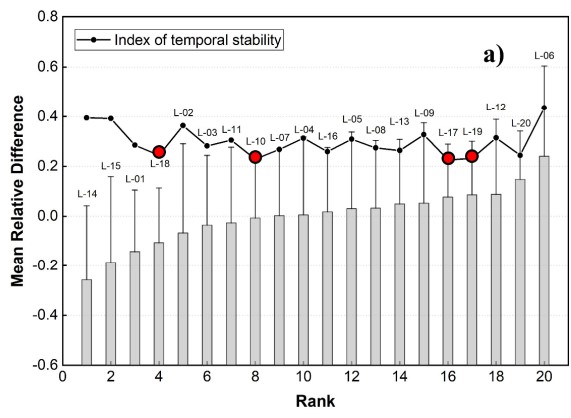
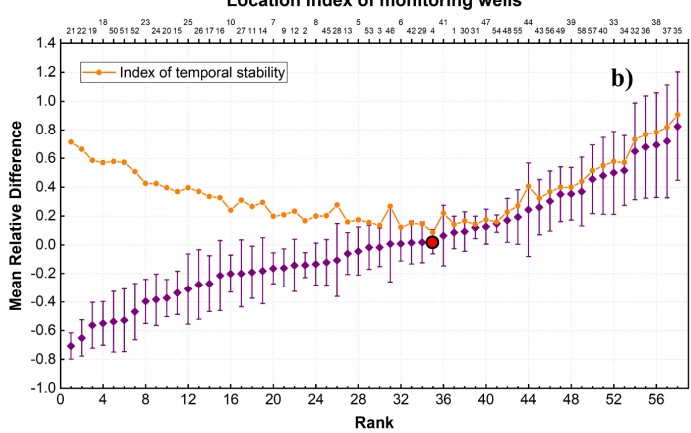


**Figure 10: (a) Rank ordered mean relative difference of soil moisture represented with the vertical bar for each sampling site (b) the ordered mean relative difference of DTGT at each observation well. The ITS for each location are shown along with the vertical bound in each case indicates the ±1 standard deviation. The red filled circles represents the most time stable location in both the figures.**

Potential representative sites are characterized by low values of $|\bar{\delta}_j|$ and $\sigma(\delta_j)$ which are combined into the index of temporal stability (ITS) (Chen et al., 2016). Evaluation of ITS is suited better to identify the representative site for each measurement location. We have therefore selected points based on the lowest value of ITS. Also, we considered the interspace between the observation wells, so that the potential "representative" sites will be well spread enough to reduce the mean spatial variation of the groundwater table throughout the CZO. In case of SM monitoring sites, ITS lies in a range between 0.23 to 0.44 (Fig.

10a) and this range is quite wide (0.09 to 0.91) for DTGT observation wells (Fig. 10b). In both cases, the ITS seems to be





within the range of $\sigma(\delta_j)$ towards the high values of $\bar{\delta}_j$. The initial ten ITS values with their increasing order are shown in Table 1 along with the corresponding observation sites/wells for both the datasets. The temporally stable measurement site(s) are addressed here by characterising the corresponding ITS values and evaluation of goodness of statistics towards the CZO mean estimates.


**Table 1: Ten most temporally stable locations (based on their ITS value) along with their corresponding mean relative difference and standard deviation of mean relative difference.**

| Soil moisture | | | | DTGT | | | |
|---|---|---|---|---|---|---|---|
| Location index | ITS | $\bar{\delta}_j$ | $\sigma(\delta_j)$ | Well index | ITS | $\bar{\delta}_j$ | $\sigma(\delta_j)$ |
| L-17 | 0.226 | 0.077 | 0.212 | W-04 | 0.09 | 0.02 | 0.08 |
| L-10 | 0.228 | -0.007 | 0.228 | W-06 | 0.12 | 0.01 | 0.12 |
| L-19 | 0.231 | 0.086 | 0.215 | W-03 | 0.14 | -0.02 | 0.13 |
| L-20 | 0.244 | 0.148 | 0.194 | W-01 | 0.14 | 0.09 | 0.11 |
| L-18 | 0.244 | -0.106 | 0.220 | W-31 | 0.14 | 0.12 | 0.08 |
| L-16 | 0.258 | 0.017 | 0.258 | W-29 | 0.14 | 0.02 | 0.14 |
| L-13 | 0.263 | 0.049 | 0.259 | W-42 | 0.15 | 0.02 | 0.15 |
| L-07 | 0.267 | 0.003 | 0.267 | W-53 | 0.15 | -0.02 | 0.15 |
| L-08 | 0.273 | 0.032 | 0.272 | W-13 | 0.16 | -0.06 | 0.15 |
| L-03 | 0.281 | -0.035 | 0.279 | W-54 | 0.16 | 0.15 | 0.06 |

**4.7.2 Single site estimation**

Fig. 11 shows the selection and evaluation metric for a single "representative" SM site as well as the DTGT observation well.

We used the determination coefficient, $R^2$, root mean square error, RMSE, and Nash–Sutcliffe efficiency coefficient, NSE, as the metric between the time series of spatial mean and measurements at the representative site itself, for evaluating the temporally stable sites. The NSE is a normalized statistic parameter widely used in hydrologic studies that determines the relative magnitude of the residual variance compared to the measured variance (Nash and Sutcliffe, 1970; Hwang et al., 2012). The NSE can be estimated as per the following equation:

$$NSE = 1 - \frac{\sum_{k=1}^{M}(\bar{\theta}_k - \bar{\theta}_{jk})^2}{\sum_{k=1}^{M}(\bar{\theta}_k - \widehat{\bar{\theta}_k})^2} \tag{17}$$

where, $\bar{\theta}_k$ is the spatial mean values for $k^{th}$ sampling day, $\widehat{\bar{\theta}_k}$ is the average value of $\bar{\theta}_k$ for all sampling days, $j$ can be a single or mean of the combined selected sites and $M$ is the total number of sampling days. The NSE can be an indicative of the goodness of fit of the observed and simulated values with 1:1 line. This means that the model is predicting well when the NSE is close to 1.





The selection of a particular site was based on the preference on the order of the lowest to highest ITS values. So, considering
the lowest value of ITS for both the datasets, the L-17 (L for SM sampling location) SM site and W-04 (W for observation
well) DTGT observation well in the CZO (Fig. 1) as the single representative site, the annual mean spatial variability can be
computed in the study region (Table 1). However, time series of the corresponding measurement values of L-17 are plotted
against the actual spatial mean and it can be seen that the goodness of fit statistics, $R^2$, RMSE and NSE are fairly low having

0.659 and 4.8% and 0.525 respectively for SM (Fig. 11). But this relation performs extremely well when a single monitoring
well (W-04) is considered which captures the mean spatial DTGT of the study region. The goodness of fit statistics ($R^2$=0.959,
RMSE=0.36 $mgbl$, NSE=0.883) agrees with the observed values (scatter plot in row 2, column 3 of Fig. 11). The reason for
the requirement of a smaller number of monitoring well might be because of the directional dependence of sub-surface water
availability in compared to SM which exhibit large spatial heterogeneity. Although, a single time stable representative site can

demonstrate the average values for a catchment, it is worth evaluating multiple combinations which might predict the overall
temporal pattern accurately (discussed in 4.7.3).

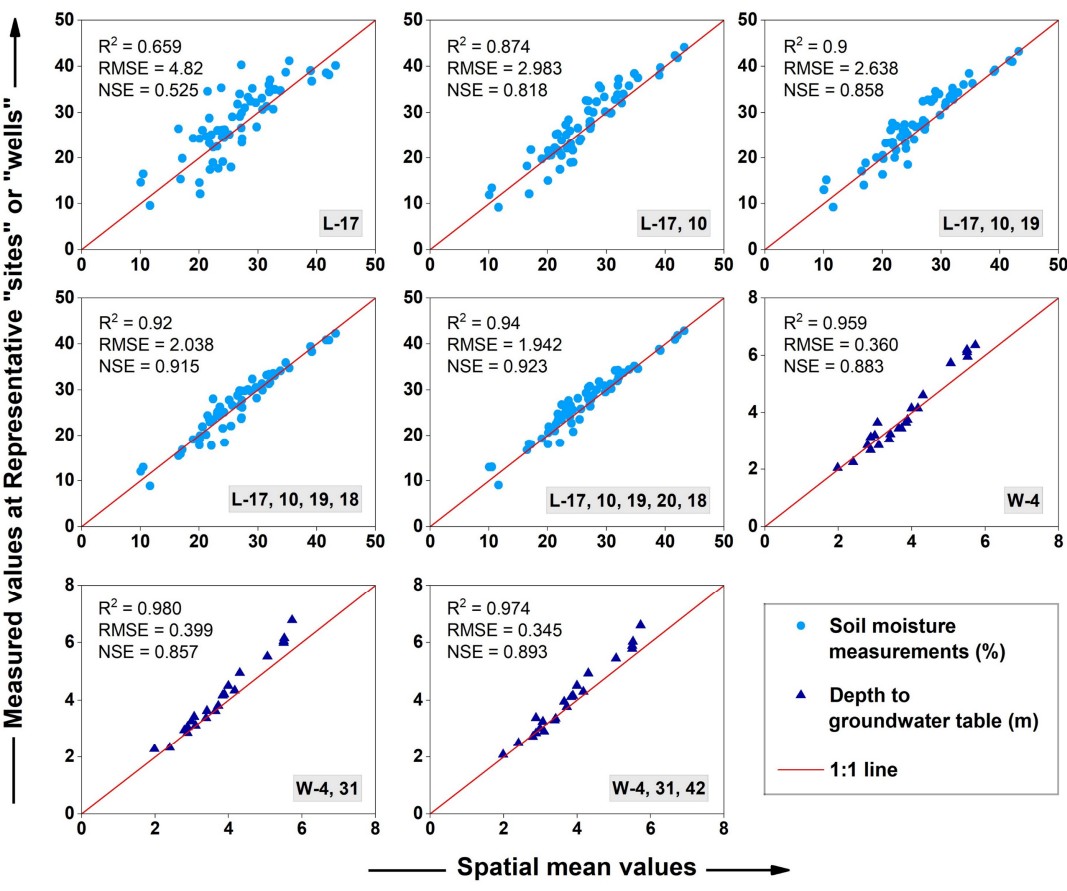

Figure 11: Scatter plots showing the comparison of CZO spatial mean value and the observed value at a single representative site or
the mean of one or more sites for both soil moisture and DTGT. The corresponding R2, RMSE and NSE values are shown for each
scatter plot. The red lines represent the 1:1 relation.



### 4.7.3 Multiple site combinations

Since all site locations behave independently because of the spatial heterogeneity that influences the existing parameters, a single time stable point (not always) is limited by reproducing measurements for all spatially located sites (Schneider et al., 2008). So, here we consider a multiple combination of sampling sites, as it might further decrease the error and thereby increase

the determination coefficient for predicting the actual spatial mean. This is also presented in Fig. 11, where selected combinations (up to five) of SM sites and observation wells (up to three) are shown, based on the increasing order of ITS. An interchangeable ITS value (0.244) is observed for the SM sampling site, L-18, and L-20 (Table 1). However, a close examination of the sampling distance between these two sites and their metrics of evaluation to approach the actual spatial mean reveals that L-17, L-10, L-19, and L-18 can be considered as the "optimal" measurement sites to capture the temporal

pattern of the spatial mean SM values in the study region with an error of ±2%. It can be seen that the goodness of fit statistics is gradually increasing from a single site to a combination of five most temporally stable sites for SM measurements. However, the combination of all the five locations does not show any significant improvement in RMSE and NSE compared to the previous four SM sites.

A similar analysis has been produced in identifying the temporally stable monitoring open well for DTGT measurements in

CZO (Fig. 11). We consider the well index 31 as the second most representative site for the DTGT measurements with the single optimal site (W-04), even though there are three intermediate open well sites (W-06, W-03, W-01; Table 1). This is because the intermediate observation wells are located close to W-04 (Fig. 1) and also it is found that the $\sigma(\delta_j)$ values of those locations are relatively higher (0.11-0.13), compared to W-31. Parameters of the goodness of fit statistics for the combined mean DTGT (W-04 and W-31) also show a good correlation with the spatial mean DTGT i.e., $R^2$=0.980,

RMSE=0.399 and NSE=0.857. Further, the $R^2$ appears to be low, when the next ranked observation well (W-42) is added to list (scatter plot in row 3, column 2 of Fig. 11). In fact, we chose the combination average of adding W-42 because the intermediate observation well (W-29, ITS = 0.14, Table 1) is closely located near to W-31 (ITS = 0.14). The performance statistics between the spatial mean DTGT and the average time series of well indices 4, 31 and 42 shows an error value of ±0.345 $mgbl$ and NSE of 0.893, which is comparatively good enough in terms of RMSE and NSE than the previous

combinations.

### 5 Discussion

### 5.1 Hydrological variability and controls of SM and DTGT over the CZO

Spatiotemporal variability of the surface soil moisture and sub-surface groundwater level has been globally reported for many hydroclimatic zones (Brocca et al., 2012; He et al., 2020; Joshi et al., 2021; Ye et al., 2019). However, most of the studies

constrain to the space-time dynamics of a single hydrological attributes. Further, the inherent control over space is extremely limited in these studies, especially for the cyclic agriculture (paddy/wheat) zones. In this study, we present an integrated





approach of EOF analysis and statistical techniques to understand the variability-control and the optimal sampling strategy over space-time utilizing the spatiotemporal SM and the DTGT observations.

The observed spatiotemporal dynamics of SM and DTGT exhibit a strong seasonality to rainfall pattern, controlling in-situ
temporal dynamics (Fig. 3). The concave upward relation between $\bar{\theta}_k$ and $CV_k$ and convex upward relationship between $\bar{\theta}_k$ and $\sigma_k$ in the SM dynamics in Fig. 5a have been observed from numerical studies across various climatic domains (Brocca et al., 2010, 2012; Peterson et al., 2019). Further, the obtained relationship is also convincing for the DTGT observations (Fig. 5b). Interestingly, the average temporal variability, $CV_j$, for both SM and the DTGT data found consistent which suggests a similar rate of variation in the space-time dynamics for both the components, although there exists an inhomogeneity in the
soil physical properties.

A distinct pattern on the SM temporal mean was noticed throughout the sampling locations. It is observed that the sampling sites that have high temporal means ($\bar{\theta}_j$) shows a low temporal variability ($\sigma_j$) and vice versa (Fig. 4). This observation was also manifested when considering the leading EOF pattern of SM and DTGT, where around 83% of the observed spatiotemporal DTGT pattern and 25% of the SM is explained by the first EOF/EC pair (Fig. 6c) and this observation is in
accordance with Yu and Lin (2015) for the groundwater study in Pingtung Plain, Taiwan. The negative values of EOF1 in the downstream reveals a drier annual SM, consistent with Singh et al., 2019 for a tropical basin in Eastern India. Conversely, the positive EOF1 pattern of DTGT in the downward side of the catchment shows a deeper water level compared to the upstream (Fig. 6a). This observation could be better represented through mapping of the seasonal (pre-monsoon and post-monsoon) variability of DTGT, and the results shows that the temporal means of DTGT at the downstream is more than 4 $mgbl$ in
compared to the upstream (<4 $mgbl$) (Fig. 12). In addition, this is attributed to a 15 km long upper Ganga canal along the eastern edge of the CZO along with more than 70% of water bodies, present in the upstream part of CZO.

Controls on the above observations are revealed in this study through correlation analysis between the major spatial EOFs and the in-situ properties. An outcome of moderate correlation of the EOF1 of SM spatial pattern with the elevation and slope suggests that the elevation has some influence on the SM variability (Fig. 7a), and this might contribute to the present spatial
variation of SM in the CZO catchment. The estimated correlation between the slope and the EOF1 for spatial SM is in accordance with Jawson and Niemann (2007) for the Southern Great Plains and is also being documented by Zhao et al. (2020c). The clay percentages in the surface soil becomes the second control of the SM spatial variability (Fig. 7a). This relationship can intuitively be true as the downstream of the catchment lacks clay fractions which has more moisture retention capacity in compared to other soil texture (Jawson and Niemann (2007); Pan and Peters-Lidard, 2008). The distribution of clay
along with silt fractions also plays a significant role in the DTGT variability. It is evident from the strong negative correlation of clay percent with the EOF1, contrary to Yue et al., 2020, that drives the overall DTGT spatial pattern, although control of topography is reflected from its correlation with EOF2 spatial pattern in the CZO (Fig. 7b).





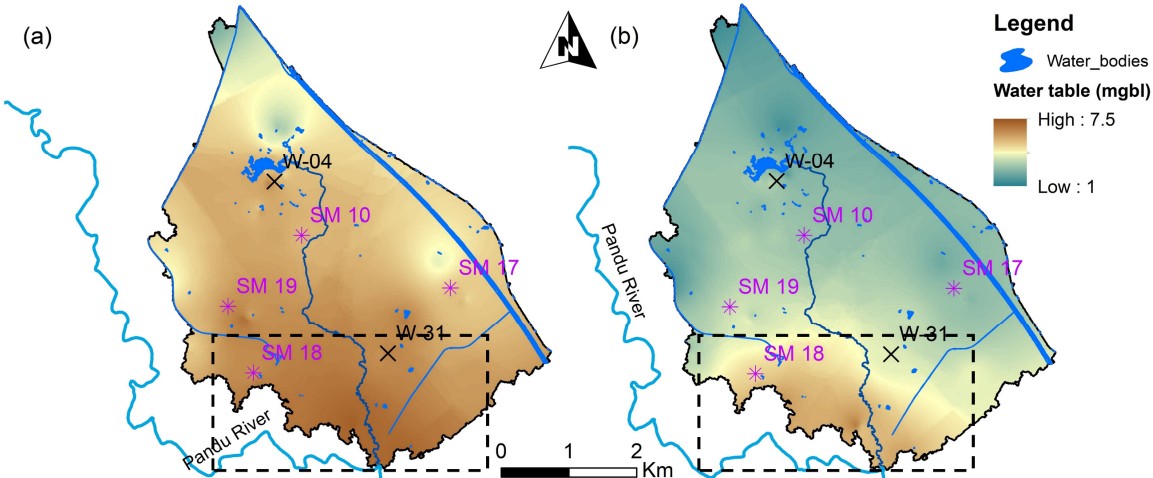

**Figure 12: Groundwater level map of CZO interpolated through inverse distance weighted (IDW) for (a) pre-monsoon and (b) post-**
**monsoon. Water bodies present in the CZO are mapped for better interpretation of the sub-surface water variability. The most**
**temporally stable open well location (W-04 and W-31 marked with black cross) and soil moisture monitoring sites (SM 17, SM 10,**
**SM 19, and SM 18 marked with purple star) are shown. Dashed box specifies the annual deeper groundwater level in the CZO.**

### 5.2 Significance of optimal measurement site(s) within CZO

Quantifying the number of samples and the site locations for estimating the mean catchment behavior was another important objective of this work, which can significantly reduce the measurement networks with a predefined accuracy. It is observed that a single random site (Fig. 8a) for the spatial mean SM produces a RMSE value which closely matches with the findings of Singh et al. (2019) for a larger scale sampling. We found that more than 10 randomly selected locations are needed for SM and 9 random observation wells are needed to estimate the corresponding spatial mean value with a very good accuracy ($R^2 \sim$ 0.98 and RMSE < 1%) respectively (Fig. 8) compared to the previous study by Gao et al. (2013), choosing four random SM sites for a similar accuracy in a large gully in the Loess Plateau of China. Likewise, according to the results on the mean relative difference of various measurement sites, characterizing it to obtain the benchmark mean DTGT and SM, the $\bar{\delta}_j$ found in this study (Fig. 10a) shows a similar range as reported by Li and Shao (2014) for an irrigated region in China and for a larger spatial extent (178 km$^2$ and 242 km$^2$) in Central Italy (Brocca et al., 2012). Although, there are very few studies that documents the time stability analysis on the DTGT measurements, we reported a strong temporal persistence (p values<0.01, Fig. 9b) here and the range of $\bar{\delta}$ (Fig. 10b), is in consistent with Wang et al. (2018), obtained for 2004-2005 groundwater depth from 18 observation wells in Yellow River Delta, China.

In this paper, the ITS along with the $\bar{\delta}_j$ was considered for quantifying the most "representative" site(s) for the observation datasets and the range of ITS found for the SM sites are in accordance with the recent studies by Dari et al. (2019) and Zhao et al. (2020b) for different climatic regions and land uses classes. We found more than one representative site for the SM datasets (Fig. 11), in consistent with Singh et al. (2019) and in contrast to previous studies (Brocca et al., 2012; He et al., 2020; Wang et al., 2013; Zhu et al., 2020). A possible explanation to the identified time-stable SM sites (L-17, 10, 19, 18) is that the





most of them are located close to the water bodies/canals which maintain an overall wetter local condition, compared to the nearby sites. Further, the selected time-invariant sites have a wide distribution of clay fractions, ranging from 25% to 28% (maximum 30% for the SM sites), where the low clay bearing crop fields significantly influence the dry period SM spatial variability (Gao et al., 2013; Jia et al., 2013). The selection of W-04, as a single stable observation well for mean DTGT is also attributed to the vicinity of this point to a large water body upstream. Selection of DTGT monitoring well is also strongly controlled by clay content (Fig. 7b) and this is reflected on the combined selection of more than one temporally stable groundwater wells (W-4, 31) where the clay distribution has a similar range e.g., SM having 25% at W-31 and 28% at W-04. Although, combined monitoring predicts better mean DTGT in this region, a single observation well is sufficient (Fig. 11).

## 5.3 Implications for irrigation water management strategies in the CZO

Knowledge on the space-time variability of the soil moisture and groundwater is invariably useful for many hydrological applications which can improve land and agricultural water management, especially in water-limited environment (Hu et al., 2019; Sekhar et al., 2016; Zhu et al., 2020). On the other hand, the in-situ SM and DTGT appear to be influenced by several natural and anthropogenic forcings (Asoka and Mishra, 2020; Omer et al., 2020; Van Loon et al., 2016), particularly in a purely agricultural domain such as the present CZO. We have integrated the information on the sub-surface architecture, our SM and DTGT observations, and local information gathered from several stakeholder engagements to represent various hydrological processes operating in the HERAT CZO (Fig. 13). The schematic diagram represents the section view of the critical zone from South-West to South-East tracking from W-37 to W-24 observation wells. Along this sequence, the DTGT varies depending upon topography, sub-surface soil texture and several pumping activities. The spatiotemporal SM and groundwater level dynamics are not only being influenced by natural surface and near-surface processes but are also modified by anthropogenic factors (Fig. 13).

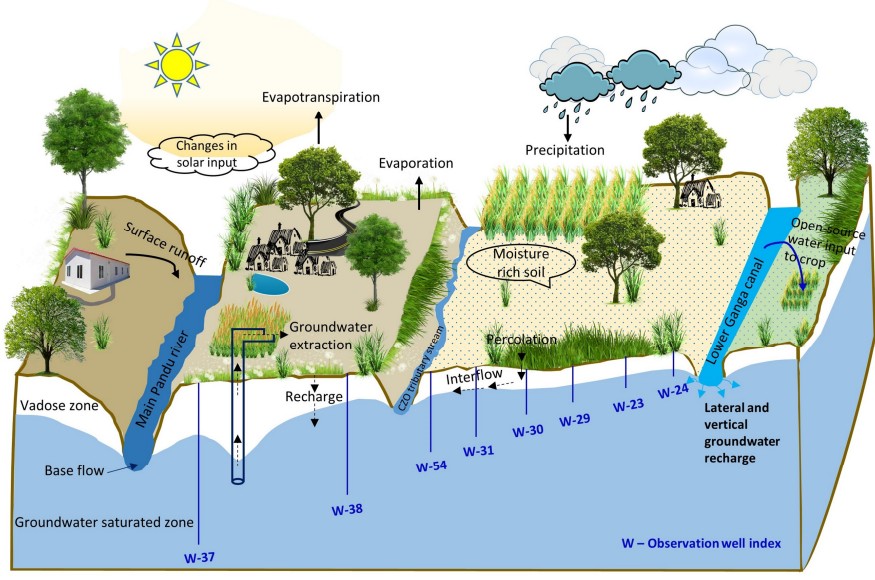



**Figure 13: Schematic representation of various hydrological processes within the study region (the HEART CZO in Ganga basin), represented with projection of a cross section tracing from south-west to north-east. The precipitation along with the canal network is supposed to be the sources of recharge in the system and contribute to the soil moisture and the groundwater through percolation. Application of water inputs to the crops are shown for the open water where the local stakeholder keeps watering the crop fields using the open sources such as the canals, river, and abandoned water bodies. Also, most of the crop fields in this CZO are fed through groundwater system where the sub-surface water extracted using the groundwater pumping system which creates the cone of depression near the point of extraction. In addition, a set of observation wells are drawn here for better representation of the subsurface water table depth at various locations. The evapotranspiration is contributed from the soil moisture and the open water bodies and is the only natural outgoing attribute in the system (Figure not to scale).**

Based on above observations and multiple discussion with the local stakeholders, following suggestions can be implemented in the CZO for efficient irrigation water management strategies:

  i)   Soil texture and topography are found to be the major controlling factors for the spatial variability in water resources (Fig. 7). Therefore, use of water resources should spatially be limited by considering specific soil characteristics. It is therefore recommended for a detail and high-resolution soil texture mapping of this region which can ultimately reduce the uninterrupted groundwater extraction.

  ii)  Correlation between individual sampling days of soil moisture in Fig. 9a suggests that the intermittent monsoon period exists up to three to four months in a year. Therefore, various water retention structures, particularly in the abandoned topographical depressions, can help in conserving rainwater for further use in the post-monsoon harvesting. Studies show that conservation of rainwater in one-tenth fallow land can support at least 60% and 75% rice and wheat, respectively (Ambast et al., 2006).

  iii) Since most of the representative SM locations/wells are close to the water bodies or the CZO mainstream (Fig. 12), an expansion/restoration of the canal network would not only reduce the overall groundwater extraction but will also maintain the threshold variability of the SM and DTGT as predicted from a limited number of sites/wells (Fig. 8 and Fig. 10).

  iv)  Typically, 4-5 hours of pumping (3–4-inch opening diameter) of groundwater for watering a 2040 sq. meter crop field (mostly wheat and rice) was noticed in parts of the CZO that are deprived of open water availability (canal/river). Indiscriminate use of groundwater for intensive agriculture leads to declining of the water table thereby causing an imbalance in the hydrological processes (Fig. 13). Therefore, apart from rainwater harvesting, field scale extraction of groundwater can possibly be reduced here by incorporating moderate water intensive crops such as mustard, gram, and other pulses in those areas.

## 6 Conclusions

The present study is the first systematic attempt to address the spatiotemporal dynamics of ground-based soil moisture (SM) and depth to groundwater table (DTGT) observations over an agriculture driven critical zone observatory (CZO) in the central Ganga plains, India. The Empirical orthogonal function (EOF) analysis to decompose the observed data into a set of spatial EOFs and temporal ECs identified that around 91% of the total DTGT spatial variance and 67% of the total SM variance are

explained by first two and five EOFs, respectively. In addition, elevation and clay percentages are observed to be the major drivers of spatial variability of both the attributes. We found a constant average temporal coefficient of variation in both SM

and DTGT, which suggests a consistent seasonal change in the surface-subsurface water dynamics. Random combination on both the observations revealed that four SM monitoring sites and two open wells for DTGT can capture the corresponding spatial mean with ± 3% ($vol/vol$) and ± 0.90 $mgbl$ respectively. Using the approach of temporal stability, a single representative open-well (at the upstream of CZO) is identified that can reproduce the spatial mean DTGT with an absolute error of 0.36 $mgbl$, determination coefficient of 0.959 and NSE of 0.883. Furthermore, a strong temporal stability was

observed in the spatial patterns of DTGT as compared to SM at the CZO scale. Additionally, four temporally stable SM sampling sites are identified which are consistent in maintaining the temporal pattern CZO mean SM variation with an absolute error of 2% ($vol/vol$), determination coefficient of 0.92, and NSE of 0.915, irrespective of the dry or wet periods. Findings of this study can not only help in understanding the surface and sub-surface hydrodynamics but can also provide important insights for designing sustainable water resource management strategies.

**Data availability.** The datasets presented in this study can be obtained upon request to the corresponding author. All other datasets cited in the text can be retrieved following the instructions in the relevant citations.

**Author contributions.** Saroj Dash contributed in terms of conceptualization, methodology, formal analysis, and original draft preparation. R. Sinha contributed in terms of conceptualization, supervision, review and editing, resources, project administration and funding acquisition. All authors have read and agreed to the published version of the manuscript.

**Competing interests.** The authors declare that they have no conflict of interest.

**Acknowledgements.** The CZO in the Ganga basin was established with the financial support from the Ministry of Earth Sciences, Government of India. This CZO was further supported by the generous funding from the STFC, UK and AvH, Germany. All these organisations are acknowledged for their support during the last 5 years. We also thank Dr. Shivam Tripathi and Surya Gupta for helping us to set up the instruments and for several fruitful discussions during the executing of this project.

We acknowledge the help of Dr. Manudeo Singh in clarifying some issues in data analysis. This work is a part of the Ph.D. work of the first author (SKD), and he acknowledges the financial support in the form of institute fellowship from IIT Kanpur.

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
