# Peer review of "Spatiotemporal dynamics and interrelationship between soil moisture and groundwater over the Critical Zone Observatory in the Central Ganga plain, North India"

_Hydrology and Earth System Sciences, 2022_

## Author Comment (AC1)

**Interactive Discussion: Author Response to Referee #1**

Spatiotemporal dynamics and interrelationship between soil moisture and groundwater over the Critical Zone Observatory in the Central Ganga plain, North India

Dash S. K. and Sinha R.,
Hydrol. Earth Syst. Sci. Discuss., 2022

Dear Referee,

We would like to thank you for your critical remarks and feedbacks on this study. Please find below our detailed responses to the points that you raised, and we hope this will fulfill the requirements of the concerns.

1. […] This a regional study that uses only existing methods, both in terms of measurement and methodology, so the results have no fundamental scientific added value outside the study area. Therefore, this work does not fit in this journal. Alternatively, there are many journals where regional studies like this can be published.

The present manuscript describes the first order attempt to use both the ground-based soil moisture (20 spatial points) and depth to groundwater table (DTGT) (58 observation well) datasets in the newly established Critical Zone Observatory (CZO, the second one in India) in the central Ganga plain, North India. This CZO is one of the very few CZOs across the world that is located in an intensively managed landscape, and therefore the findings from this paper have significant implications to advance the critical zone science, which is a fast-emerging topic in earth surface processes (Brantley et al., 2017; Gaillardet et al., 2018; Guo and Lin, 2016; Li et al., 2018). Further, the present CZO was established to understand the baseline conditions and sub-surface hydrodynamics, as a representative of the Ganga basin in North India (Gupta et al., 2019), which is considered as the 'food basket' of India. Therefore, outcomes of this study would be highly beneficial for designing the sampling strategies and understanding the primary control on the major variability of the hydrologic components outside the CZO network. In addition, the selection of an optimal number of points would not only provides a baseline for short-term remote sensing evaluation studies and upscaling sensor network but also help in managing the concurrent water resources for effective management. In view of the concerns from the reviewer, we would be happy to revise manuscript accordingly to emphasize these points.

2. The temporal resolution of the data is rather poor. This is outdated nowadays, when sensors and data loggers can be acquired and operated for relatively little money.

We agree with your concerns regarding the sensor and data logger based measurements for the spatiotemporal dynamics. However, we would like to highlight that numerous recent studies have

documented such analysis with limited number of space-time networks for demonstration of interdependent observational variability (Dari et al., 2019; He et al., 2020; Li and Shao, 2014; Zhang et al., 2016). In fact, we are continuing the observations in the study region, which could potentially supplement the ground-based measurements in the Ganga basin in near future. It is also worth pointing out that installation of automated sensors and data loggers in a rural but densely populated and intensively agriculture area is not a trivial issue, and there are serious problems with safety of the sensors and network connectivity. So, the manual data collection in such areas works best to capture the temporal variability.

Another important aspect here is that we represent the observational data based on two categories, (1) spatially limited and temporally dense and (2) temporally limited and spatially dense. The first category corresponds to the surface soil moisture observations (20 location and 62 days) whereas the second category include the DTGT observation (58 well and 22 days). The novelty here is that even though the temporal resolution of the groundwater level data is relatively less, results of this study comprehensively represent the spatial variability of the groundwater dynamics as suggested by EOF analysis. Our analysis has also identified a single representative monitoring well in the CZO based on temporal stability analysis. Furthermore, the results of statistical analysis of soil moisture including random combination and temporal stability are found to be coherent with previous studies (please see Section 5.2 of the manuscript).

3. The evaluation of groundwater levels with statistical methods is problematic because groundwater level measurements are usually not independent of each other, since they observe the same groundwater body. Unless the measurements took place in clearly delimited aquifers, but this was not explained in the text.

Thank you for this thoughtful comment and careful interpretation. Yes, you are correct on the interpretation of statistical analysis for a single groundwater body. We missed to highlight this point in our manuscript about the groundwater level measurements of various delimited aquifers of this region. We are monitoring water level for the shallow aquifers ($< 8\ mbgl$) in an interfluve region of the Ganga plains. Previous studies have clearly established that these aquifers exist as narrow sand ribbons in the subsurface (please see Figure 4 and 5 of Yadav et al., 2010), and therefore are clearly delimited. The study by Yadav et al. (2010) was based on a large number of resistivity soundings across the Ganga-Yamuna interfluve region which covered our study area as well (near IIT Kanpur to the east of Pandu River). Resistivity surveys supplemented with litholog data show the presence of laterally disconnected narrow aquifer sand bodies in shallow sub-surface whereas an extensive sand body exists only at a deeper level (Yadav et al., 2010). Therefore, our statistical analysis holds good for this setting. Thank you again for raising this point, we will add this information in the revised manuscript.

4. The statement that topography and the clay content of the soils are considered the most important factors determining the spatial pattern is not tenable, as the study area consists of

irrigated agricultural fields. Therefore, the spatial pattern will strongly depend on the amount of water applied locally and the crops. Without taking this into account, the general statements cannot be made.

Thank you again for your concern. Yes, we agree with your statement regarding the control of watering and crop type on spatial dynamics of soil moisture and groundwater. The study region typically belongs to a monsoon–fed irrigation system with primary dependence of monsoonal precipitation for rice crops. However, the wheat crop in post-monsoon is typically irrigated through periodic wetting and drying through local watering, based on soil moisture assessment by the farmers. Therefore, as a concern of the #Reviewer-1, we segregated the entire soil moisture sampling window into two parts, rice crop cycle (Mid-June–October) and wheat crop cycle (November–April) and subsequently analyzed their spatial EOFs correlation with time-invariant properties (please see the figure below).

[Figure]

**Figure:** Relationship between the first five spatial EOFs of soil moisture with the time-invariant in-situ physical properties for the rice cropping cycle (upper) and for the wheat cropping cycle (lower).

As the figure shows, clay and topography followed by silt fraction are the prime factors for the spatial variability of soil moisture during rice crop cycle, whereas these factors are not able to control during wheat cycle, when the watering is periodic. In the wheat crop cycle, sand (%) and slope have a moderate correlation with EOF1, suggesting partial influence on the spatial SM variability. However, a mixed control of topography, clay (%), sand and slope contribute to most of the spatial SM variability during wheat cycle. This is visible here with a satisfactory correlation of EOF1 and EOF2 with the above factors.

We would also like to highlight that the soil moisture measurements were usually avoided during local crop watering. Additionally, SM measurements are conducted with a weekly sampling interval, therefore, actual watering day might be insignificant to the mean SM variability.

We would be happy to include this information, supplement with figures in the revised manuscript to address the Reviewer's concern.

5. Finally, strategies for efficient irrigation water management are proposed that not based on the statistical analyses in this paper.

We understand the concerns of the reviewer, but we beg to differ. In this study, we have made an effort to combine several stakeholder surveys to the understand SM and DTGT spatiotemporal variability, although the analysis includes primarily the natural hydrometeorological components. We have not made any specific recommendations, but this study has strong implications for developing future monitoring and sustainable irrigation strategies. Apart from providing guidance for selection of optimum network for soil moisture and groundwater monitoring, we found a significant correlation of monsoonal variability of soil moisture between multiple annual cycles (please see Figure 9a of the manuscript). This information can potentially be used to harvest rainwater to the abandoned/unused local depressions in the study region, which will ultimately lead to efficient irrigation water management. Further, our findings on the identification of representative time stable monitoring locations, adjacent mostly to the presence of water bodies (please refer to Figure 12 of the manuscript), would help further in restoring several unused channel network and canal to maintain a threshold soil water content in the crop field. We also recommend to minimize extensive groundwater extraction by planting moderate water intensive crops depending upon the textural characteristics of soil which primarily governs the groundwater variability (please see figure in #comment-4 above). We would be happy to expand this section a bit more to highlight these points more explicitly.

References

Brantley, S.L., McDowell, W.H., Dietrich, W.E., White, T.S., Kumar, P., Anderson, S.P., Chorover, J., Lohse, K.A., Bales, R.C., Richter, D.D. and Grant, G., 2017. Designing a network of critical zone

observatories to explore the living skin of the terrestrial Earth. Earth Surface Dynamics, 5(4), pp.841-860. https://doi.org/10.5194/esurf-5-841-2017

Dari, J., Morbidelli, R., Saltalippi, C., Massari, C. and Brocca, L., 2019. Spatial-temporal variability of soil moisture: Addressing the monitoring at the catchment scale. Journal of Hydrology, 570, pp.436-444. https://doi.org/10.1016/j.jhydrol.2019.01.014

Gaillardet, J., Braud, I., Gandois, L., Probst, A., Probst, J.L., Sanchez-Pérez, J.M. and Simeoni-Sauvage, S., 2018. OZCAR: The French network of critical zone observatories. Vadose Zone Journal, 17(1), pp.1-24. https://doi.org/10.2136/vzj2018.04.0067

Guo, L. and Lin, H., 2016. Critical zone research and observatories: Current status and future perspectives. Vadose Zone Journal, 15(9). https://doi.org/10.2136/vzj2016.06.0050

Gupta, S., Karumanchi, S.H., Dash, S.K., Adla, S., Tripathi, S., Sinha, R., Paul, D. and Sen, I.S., 2019. Monitoring ecosystem health in India's food basket, Eos, 100. https://doi.org/10.1029/2019EO117683

He, M., Wang, Y., Wang, L. and Li, R., 2020. Spatio-temporal variability of multi-layer soil water at a hillslope scale in the critical zone of the Chinese Loess Plateau. Hydrological Processes, 34(23), pp.4473-4486. https://doi.org/10.1002/hyp.13888

Li, D. and Shao, M.A., 2014. Temporal stability analysis for estimating spatial mean soil water storage and deep percolation in irrigated maize crops. Agricultural Water Management, 144, pp.140-149. https://doi.org/10.1016/j.agwat.2014.05.012

Li, X., Yang, X., Ma, Y., Hu, G., Hu, X., Wu, X., Wang, P., Huang, Y., Cui, B. and Wei, J., 2018. Qinghai Lake Basin Critical Zone Observatory on the Qinghai-Tibet Plateau. Vadose Zone Journal, 17(1), pp.1-11. https://doi.org/10.2136/vzj2018.04.0069

Yadav, G.S., Dasgupta, A.S., Sinha, R., Lal, T., Srivastava, K.M. and Singh, S.K., 2010. Shallow sub-surface stratigraphy of interfluves inferred from vertical electric soundings in western Ganga plains, India. Quaternary International, 227(2), pp.104-115. https://doi.org/10.1016/j.quaint.2010.05.030

Zhang, Y., Xiao, Q. and Huang, M., 2016. Temporal stability analysis identifies soil water relations under different land use types in an oasis agroforestry ecosystem. Geoderma, 271, pp.150-160. https://doi.org/10.1016/j.geoderma.2016.02.023

---

## Author Comment (AC2)

**Interactive Discussion: Author Response to Referee #2**

Spatiotemporal dynamics and interrelationship between soil moisture and groundwater over the Critical Zone Observatory in the Central Ganga plain, North India

Dash S. K. and Sinha R.,
Hydrol. Earth Syst. Sci. Discuss., 2022

Dear Referee,

We appreciate for your time to review our manuscript and providing your thoughtful comments and suggestions. Please find below our detailed responses to your concerns.

Based on multi-year observations of soil moisture and groundwater table at spatially distributed sites over an agriculture-driven critical zone observatory, the authors investigate the spatiotemporal dynamics of both variables, using a series of analysis methods (incl. EOF, random combination, temporal stability analysis, etc). Based on the findings of these analyses combined with stakeholders' surveys, some water management strategies were proposed. The paper is very well written, structured, and clear.

Thank you for your constructive comments and feedback. We hope that the analysis carried out in this study would help in designing sampling strategies for water resource management.

**Below please find some main concerns:**

1. Although the analysis carried out is very convincing in terms of finding representative sites/wells for understanding the spatial mean of the CZO, it is however not clear how the human activities (irrigation/groundwater extraction) will impact such analysis. There were several places the author indicate some spatiotemporal patterns to irrigation and pre-monsoon, post-monsoon precipitations. However, this information cannot be explicitly found in the manuscript. Please the authors try to clarify this information and explain their potential impacts on their findings.

We thank the reviewer for bringing up this point. To understand the control of irrigation watering on spatiotemporal control of soil moisture, we segregated the entire temporal window into rice and wheat crop cycle. This is because the farmers in the study region primarily grow rice in monsoon and wheat during non-monsoon months with limited summer crops. The rice cropping is usually dependent on the atmospheric precipitation whereas non-monsoon month crops are typically irrigated through periodic wetting and drying through local watering, based on soil moisture assessment by the farmers. The EOF analysis (Figure below) for each cycle shows that clay and topography remain as prime driver of SM spatial dynamics in rice cycle, when natural variability contributes most. But SM spatial dynamics during the wheat crop cycle is controlled with a

combined effect of topography–clay and sand–slope. These are shown in the figure below as correlations of first few EOFs with corresponding physical properties. Considering the irrigation scheduling to be a representative of the central Ganga plain, our findings on representative sites for rice and wheat cycles are the salient outcomes of this study.

[Figure]

**Figure:** Relationship between the first five spatial EOFs of soil moisture with the time-invariant in-situ physical properties for the rice cropping cycle (upper) and for the wheat cropping cycle (lower).

Secondly, for groundwater observations, although effect of any seasonal or timely pumping activities influence its temporal pattern, any specific variation due to pumping activities is apparently unnoticeable on its spatiotemporal dynamics (Figure-3b of the manuscript). This is because the mean interannual variation of groundwater level within the CZO strongly follows with natural seasonal forcings (please see Figure-3b of the manuscript) and we don't observe any trend of groundwater decline due to human impact within its study period. This suggests that extraction of groundwater is currently balanced through recharge within this region.

We also have taken care of the sampling of both components (SM and DTGT), not to be simultaneous with any of the pumping/watering activities. Also, the sampling intervals for SM and

DTGT are mostly weekly and bi-weekly respectively, so that any specific watering activities become minimal during the sampling. We hope this could potentially address the Reviewer's concern and we will revise the manuscript accordingly to include this information.

2. The satellite data SMAP was mentioned in the manuscript. However, there is no satellite SM data used in this study. This reviewer thinks this is a miss of the opportunity. It would be great to link the in-situ measurement to remote sensing data, as such, it is more operational in a sense to monitor the impact of water management strategies on soil moisture, or even groundwater storage change. It is understandable that for GW storage, the current GRACE product is too coarse. However, for SMAP soil moisture data, you do can find 1km and 3km resolution products. Also from Sentinel-1 SM, it is 1km. As such, this reviewer would encourage the author to include satellite data in their analysis.

Thank you for your encouraging suggestion on linking our ground measurements to remote sensing data. We would like to mention that the ground-based measurements on the soil moisture were mostly planned as per the revisit time of SMAP satellite. However, a detailed evaluation of the SMAP soil moisture products vis-a-vis in-situ datasets has just been published by us for the study area (Dash and Sinha, 2022, Remote Sensing, 14, 1629; https://doi.org/10.3390/rs14071629). Therefore, we have, made an effort to characterize the CZO hydrodynamics in this paper based on only the ground-based available spatiotemporal datasets. We propose to add the following information in the revised version:

*"Selection of the representative site(s) can be used for remote sensing evaluation studies or upscaling the soil moisture network (Schneider et al., 2008), more specifically during the wet season. Recently, the present network has been used for the evaluation of passive remote sensing SMAP soil moisture products, reporting a high accuracy level and minimal random error in comparison to C and X-band microwave SM products based on the annual and seasonal scale variability (Dash and Sinha, 2022)"*

**Some minor comments as below:**

a. On page 10, line 230, this reviewer is wondering if you have the data about 'watering by farmers'?

Thank you for your query. The study region, which could potentially represent the overall agricultural pattern in the Ganga plain, typically uses 4–5 hours of groundwater pumping (3–4-inch diameter pipe) for a 2040 sq. meter crop field (information from community surveys). The pumping interval is usually 3–4 times (mostly on a monthly interval) during a wheat crop cycle whereas the rice crop significantly depends on monsoonal rainfall with around 7–8 times watering for an adverse monsoon period. We would be happy to add this information in the revised version.

b. Page 10, line 234-235, this reviewer think this is only happening when the GW table is shallow, right? Please clarify and provide some more discussions on this.

Thank you for your thoughtful interpretation. You are correct that our data represents shallow groundwater table ($< 8\ mbgl$). The characteristic shallow aquifer in this interfluve region exist as narrow sand ribbons in the subsurface (please see Figure 4 and 5 of Yadav et al., 2010), and therefore are clearly delimited. Here we would also like to highlight that the $CV_j$ stands for temporal coefficient of variation and we noticed the mean value (0.36) is same for both the SM and DTGT. This suggests that although, the spatial variability of both observations is different (SM is highly heterogenous compared to DTGT), the annual temporal mean variability is consistently interdependent. We will add this information for clarification in the revised manuscript.

c. Line 265 'PC' should be 'EC'

Thank you for noticing this. We will correct this accordingly.

d. Line 338, it would be convenient for readers if equations were given.

Thank you for your suggestion. We will add the Spearman's correlation equation in the revised manuscript as follows:

*"To analyze the similarity in the spatial mean between the sampling days, the Spearman rank correlation coefficients, R, has been computed and presented in Fig. 9a and 9b for SM and DTGT, respectively. The Spearman's correlation (R) was obtained based on the two ranked variables according to the following expression:*

$$R = 1 - \frac{6 \sum_{i-1}^{n} \left( R_{X_i} - R_{Y_i} \right)^2}{n(n^2 - 1)}$$

*where $R_{X_i}$ and $R_{Y_i}$ are the ranked variables of $X_i$ and $Y_i$ ($i = 1, 2, 3, \dots, n$), respectively, and $n$ is the total number of elements for each variable. A rank correlation close to 1 indicates a stronger tendency of similarity between the variables."*

e. Line 344, these 'signals' should be marked out explicitly in Figure 9a

Thank you for your suggestion. These signals are now marked explicitly as rectangular box in the revised figure (as shown below). We have also marked the two arrows in the y-axis (revised figure here) to support the statements in Line-344. In addition, we also renamed the axis labels as 'Day counts' of a year for better clarity.

[Figure]

**Figure:** Spearman rank correlation coefficient of the observed values during the measurement campaigns of a) soil moisture, where the x and y-axis represents the day of the year. The box represents negative correlations of 11th Jan 2019 soil moisture with 2017 monsoon days. Also, the two arrows are shown which represents the 2019 November samplings with the monsoon of previous years. The correlation coefficients are shown as the triangle where the value is represented as generic boxes identified by $i_{th}$ row and $j_{th}$ column and each of the row/column are the sampling dates of the corresponding field campaign. * Indicates significance at $p < 0.05$, ** Indicates significance at $p < 0.01$

f. Line 414, 'in compared to' should be 'in comparison to'

Thank you for your suggestion. We will replace this accordingly.

References

1. Dash, S.K. and Sinha, R., 2022. A Comprehensive Evaluation of Gridded L-, C-, and X-Band Microwave Soil Moisture Product over the CZO in the Central Ganga Plains, India. Remote Sensing, 14(7), p.1629. https://doi.org/10.3390/rs14071629
2. Schneider, K., Huisman, J.A., Breuer, L., Zhao, Y. and Frede, H.G., 2008. Temporal stability of soil moisture in various semiarid steppe ecosystems and its application in remote sensing. Journal of Hydrology, 359(1-2), pp.16-29. https://doi.org/10.1016/j.jhydrol.2008.06.016
3. Yadav, G.S., Dasgupta, A.S., Sinha, R., Lal, T., Srivastava, K.M. and Singh, S.K., 2010. Shallow sub-surface stratigraphy of interfluves inferred from vertical electric soundings in western Ganga plains, India. Quaternary International, 227(2), pp.104-115. https://doi.org/10.1016/j.quaint.2010.05.030